# A viral APOBEC3 antagonist distinguishes HHV-6A from HHV-6B

Jun Arii [1,2,7] ✉, Salma Aktar [1,3,4,7], Jing Rin Huang[1], Mansaku Hirai [1], Yoshiki Kawamura [5,6], Hiroki Miura[5], Bochao Wang[1], Satoshi Nagamata[1], Mitsuhiro Nishimura[1], Tetsushi Yoshikawa[5], Reuben S. Harris [3,4] & Yasuko Mori [1]

Human herpesviruses exhibit diverse pathogenic outcomes and the molecular reasons are not fully understood. Human herpesvirus 6B (HHV-6B) causes exanthema subitum and encephalitis, whereas the closely related HHV-6A is typically asymptomatic. Here, we show that cellular APOBEC3 enzymes restrict HHV-6A replication but not HHV-6B. HHV-6B expresses higher levels of the U28 protein, which binds multiple APOBEC3 proteins and promotes their relocalization and degradation. In contrast, HHV-6A fails to counteract APOBEC3 activity and accumulates extensive mutations in both cell- and patient-derived viral genomes. Individual APOBEC3 gene ablation enhances HHV-6A replication and reduces the viral mutation burden. Together, our studies suggest that differential susceptibility to APOBEC3 restriction may help to shape the evolvability and clinical manifestations of HHV-6A and HHV-6B.

Many human viruses cause diseases with overt pathogenesis, whereas others establish persistent or chronic infections with minimal symptoms. Virus–receptor interactions, the efficiency of viral genome replication, and evasion of host immune responses are key determinants of tissue tropism and pathogenicity[1–3]. However, these factors alone fail to fully explain the distinct clinical outcomes among related viruses.

Human herpesviruses are widespread, infecting more than 90% of the adult population[4]. They persist for life by establishing latency and periodically reactivating, sometimes leading to disease[4]. Among them, human herpesviruses 6A (HHV-6A) and 6B (HHV-6B) are T lymphotropic β-herpesviruses with over 90% genomic similarity. Primary HHV-6B infection causes roseola infantum, also known as exanthema subitum or sixth disease, and its reactivation can occasionally lead to complications such as encephalitis and pneumonia, particularly in patients receiving allogeneic hematopoietic stem cell transplantation (allo-HSCT) or chimeric antigen receptor (CAR) T cell therapy[5–10]. In contrast, HHV-6A is generally asymptomatic in humans[5,6]. Despite their high sequence similarity, the molecular basis for these striking differences in pathogenic potential remains poorly understood. Subtle differences in viral tropism have been reported in cell culture[5], but their mechanistic basis is unclear.

HHV-6A is commonly propagated in CD4[+] T cell lines such as HSB2 and JJhan, which lack expression of CD134, the entry receptor for HHV-6B, and therefore cannot support HHV-6B replication[5,6,11]. Ectopic expression of CD134 renders these cells susceptible to HHV-6B infection[11]. Conversely, MT4 cells endogenously express both CD134 and CD46—the respective entry receptors for HHV-6B and HHV-6A—but paradoxically support only HHV-6B replication (Fig. 1a)[5,6,12]. Although CD46 is broadly expressed in many human tissues, most cell types remain non-permissive to HHV-6A replication[5,12]. These observations suggest that entry receptor usage alone does not dictate host cell permissiveness and that additional post-entry barriers may help to limit viral replication. We hypothesize that host intrinsic restriction factors may contribute to this post-entry blockade and thus help explain the divergent replication properties of HHV-6A and HHV-6B.

[1]Division of Clinical Virology, Center for Infectious Diseases, Kobe University Graduate School of Medicine, Kobe, Japan. [2]Graduate School of Biomedical and Health Sciences, Hiroshima University, Hiroshima, Japan. [3]Department of Biochemistry and Structural Biology, University of Texas San Antonio, San Antonio, TX, USA. [4]Howard Hughes Medical Institute, University of Texas San Antonio, San Antonio, TX, USA. [5]Department of Pediatrics, Fujita Health University School of Medicine, Toyoake, Japan. [6]Department of Pediatrics, Fujita Health University Okazaki Medical Center, Okazaki, Japan. [7]These authors contributed equally: Jun Arii, Salma Aktar. ✉e-mail: jarii@med.kobe-u.ac.jp

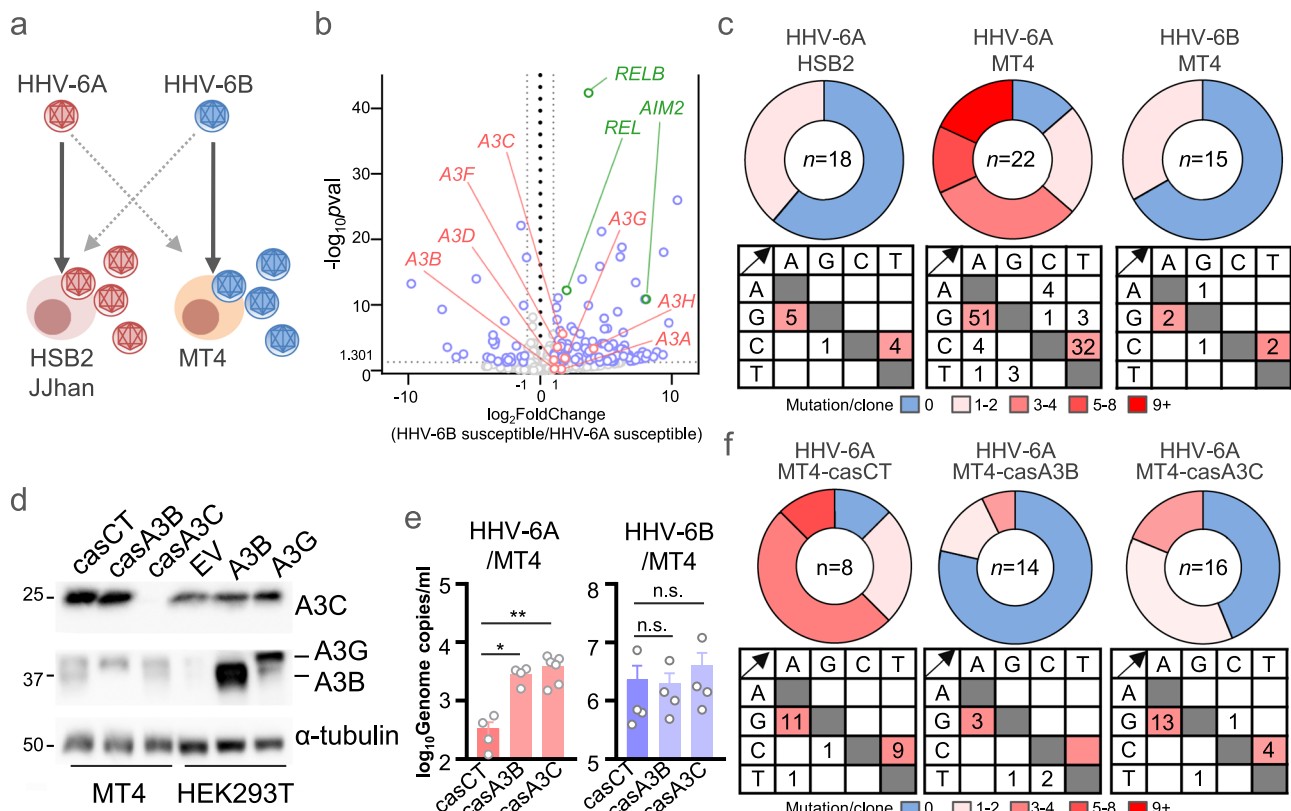

**Fig. 1 | Restriction of HHV-6A but not HHV-6B replication by APOBEC3 proteins. a** HSB2 and JJhan cells support HHV-6A but not HHV-6B replication, whereas MT4 cells support HHV-6B only. **b** Volcano plot of differentially expressed genes (DEGs) comparing HHV-6B-susceptible and HHV-6A-susceptible cells. APOBEC3 genes are labeled in pink; other known restriction factors in green; remaining innate-immune DEGs in blue, and non-DEGs in gray. Only innate-immune genes are shown. Differential expression was analyzed using DESeq2 with the Wald test; dashed lines indicate |log₂-fold change| = 1 and P = 0.05. Genes were displayed using |log₂-fold change| > 1 and unadjusted P < 0.05. **c** Pie charts and mutation matrices showing types of mutational events observed in Sanger sequences of cloned *U4* gene segments (685 base pairs) from HSB2 or MT4 cells infected with HHV-6A or HHV-6B. Non-mutated sequences are depicted in blue; base substitutions with the number of mutations per sequence are depicted in red. **d** Immunoblot of lysates from MT4 cells expressing Cas9 and the indicated gRNA (CT, control; A3B, A3B-targeting; A3C, A3C-targeting). Molecular weights are indicated in kDa. Representative of three independent experiments. **e** Viral genome copy numbers in the supernatant of MT4 cells expressing Cas9 and the indicated gRNA after infection with HHV-6A or HHV-6B for 72 h. Data are shown as mean ± s.e.m. from independent biological replicates (n = 4 for HHV-6A/CT, HHV-6A/A3B, HHV-6B/CT, HHV-6B/A3B, and HHV-6B/A3C; n = 6 for HHV-6A/A3C), where each n represents one independent infection experiment. Statistical significance was assessed separately for HHV-6A and HHV-6B using one-way ANOVA followed by Tukey's multiple-comparisons test; n.s., not significant; *P < 0.05; **P < 0.01. Exact P values and detailed statistics are provided in the Source Data file. **f** Pie charts and mutation matrices showing types of mutational events from the MT4 cells described in (**e**). Non-mutated sequences are depicted in blue; base substitutions with the number of mutations per sequence are depicted in red. Detailed statistics for (**e**) and source data for (**b**–**f**) are provided in the Source Data file.

To address this gap, we sought to define the post-entry determinants of HHV-6A and HHV-6B replication by comparing their intracellular fates in multiple T cell lines. Our findings reveal a previously unappreciated role for cellular APOBEC3 enzymes in shaping the differential replication efficiency and tropism of these closely related viruses.

## HHV-6B evades restriction by APOBEC3 proteins

To explore mechanisms that may limit HHV-6A replication in non-permissive cells, RNA-seq was performed for HSB2, JJhan, and MT4 cell lines as well as primary umbilical cord blood mononuclear cells (CBMCs), which support the replication of both viruses[5]. This analysis identified multiple differentially expressed genes with potential roles in herpesvirus replication or restriction, such as *RELB*, *REL*, and *AIM2*[13–15], which warrant future investigation. We reasoned that intrinsic restriction can be mediated by multigene modules with overlapping functions; accordingly, coordinated, family-level expression patterns could be informative even when individual genes did not reach significance. Given the exploratory nature of the RNA-seq analysis, DEGs were used to generate hypotheses; we therefore prioritized a gene family showing an overall rightward

shift of several members in Fig. 1b rather than single-gene p-values. Consistent with this prioritization, APOBEC3 family members tended to be more highly expressed in HHV-6B-susceptible cell lines than in HHV-6A-susceptible ones, even when differences did not reach statistical significance (Fig. 1b and Supplementary Fig. 1a). For example, *APOBEC3B* (apolipoprotein B mRNA editing enzyme, catalytic polypeptide-like 3B; *A3B*) and *APOBEC3C* (*A3C*) were more highly expressed in MT4 cells than in HHV-6A-permissive cell lines HSB2 and JJhan (Supplementary Fig. 1b). Of these deaminase family members, *A3C*, *A3D*, *A3F*, and *A3H* exhibited statistically significant differences (p < 0.05), whereas *A3A* and *A3B* did not (p > 0.05) (Fig. 1b). APOBEC3 proteins mediate innate immunity against different DNA viruses by catalyzing C to U deamination, which results in C/G-to-T/A mutations[16,17]. Humans have seven APOBEC3 family members (A3A, A3B, A3C, A3D, A3F, A3G, and A3H), which differ in expression profiles, subcellular localization, and substrate specificity[3,16]. Because A3B predominantly localizes to the nucleus and A3C is partially nuclear—the site of herpesvirus genome replication[4]—both may contribute to restricting HHV-6A replication. Another nucleus-penetrating APOBEC3 member, A3A, was not expressed in any of the tested cells (Supplementary Fig. 1b).

To test whether HHV-6A replication is inhibited by APOBEC3-mediated mutagenesis, we infected MT4 cells with HHV-6A or HHV-6B for 72 h. The viral *U4* gene was chosen for sequence analysis due to its high GC content, which may make it more susceptible to APOBEC3 deamination. Sequencing of individual PCR clones revealed extensive APOBEC3 signature mutations in the HHV-6A *U4* gene in infected MT4 cells (C/G-to-T/A mutations in >80% of sequences), whereas most sequences from HHV-6B-infected MT4 cells remained wild-type (C/G-to-T/A mutations in <30% of sequences; Fig. 1c and Supplementary Fig. 2a). Consistent with APOBEC3 mutagenesis, these substitutions were enriched in the TCW (W = A/T) trinucleotide context and exhibited strand asymmetry (Supplementary Fig. 2b), features typical of APOBEC3-associated single base substitution (SBS) mutation signatures (SBS2)[18,19]. Because context analysis alone does not distinguish A3A, A3B, and A3C, we refrain from assigning a single enzyme; however, the bias away from CpC motifs makes A3G unlikely[20].

To assess the role of individual APOBEC3 proteins, MT4 cells were transduced with lentiviral vectors expressing Cas9 and a gRNA targeting *A3B* or *A3C* (Fig. 1d). This approach was efficient, as evidenced by minimal A3B and A3C expression by immunoblotting, without affecting other APOBEC3 proteins. Following infection, knockout of *A3B* or *A3C* significantly increased HHV-6A replication but had no effect on HHV-6B replication, as measured by qPCR (Fig. 1e). Additionally, HHV-6A genomes in *A3B*- or *A3C*-depleted cells exhibited fewer mutations compared to those from MT4 cells expressing Cas9 alone (Fig. 1f). These results indicate that HHV-6A is susceptible to mutagenesis by A3B and A3C, whereas HHV-6B escapes restriction.

## HHV-6B U28 redistributes and degrades APOBEC3 proteins

Viruses have evolved different mechanisms to counteract the activity of APOBEC3 proteins[3,16]. A well-characterized example is the human immunodeficiency virus type 1 (HIV-1) Vif protein, which targets multiple APOBEC3 enzymes, including A3G, for proteasomal degradation[3,16,21,22]. More recently, the large subunit of the ribonucleotide reductase (RNR) encoded by the γ-herpesvirus Epstein-Barr virus (EBV), BORF2, was shown to inhibit A3B enzymatic activity and relocalize A3B to the cytoplasm[23–25]. Although RNR large subunits are conserved across *Herpesviridae*, those encoded by human β-herpesviruses, including HHV-6, have lost both enzymatic activity and the ability to inhibit programmed cell death[26,27].

Whereas infection by other herpesviruses excludes A3A and A3B from the nucleus to the cytoplasm[23,24,28], HHV-6B infection led to nuclear exclusion of A3B and A3C in MT4 cells (Fig. 2a, b). Different from those, HHV-6B infection of MT4 cells not only promoted redistribution of APOBEC3 proteins but also reduced their overall protein levels (Fig. 2c, d), suggesting that HHV-6B actively counteracts these host restriction factors.

To investigate the viral factor responsible for this effect, we focused on the HHV-6B gene *U28*, which encodes a catalytically inactive RNR large subunit. HHV-6B *U28* is classified as an early gene, expressed during viral DNA replication. The U28 protein has been reported to inhibit host NF-κB signaling[29].

To examine whether HHV-6B U28 interacts with and modulates APOBEC3 proteins, we conducted three complementary experiments. First, we co-transfected HEK293T cells with constructs expressing individual APOBEC3 proteins fused to AcGFP (A3–AcGFPs) and HHV-6B U28 fused to TagRFP (TagRFP–HHV-6B U28) and analyzed their subcellular localization by confocal microscopy. We observed that most A3–AcGFPs relocalized to the cytoplasm and colocalized with TagRFP–HHV-6B U28 (Fig. 3a, b). The only exception was A3H–AcGFP, whose localization was unaffected by co-expression of TagRFP–HHV-6B U28. To further clarify the nature of these cytoplasmic structures, we performed line-intensity profiling of fluorescence signals. This analysis revealed overlapping intensity peaks for U28 and several APOBEC3 proteins, including A3B, A3C, and A3G, supporting the interpretation that they colocalize within defined cytoplasmic domains (Supplementary Fig. 3). In contrast, A3H and AcGFP did not show such overlap or punctate cytoplasmic distribution. These findings strengthen the conclusion that HHV-6B U28 selectively recruits and redistributes specific APOBEC3 proteins to the cytoplasm (Fig. 3c). Second, immunoblotting of HEK293T cells transfected with A3–AcGFP and Flag–HHV-6B U28 showed reduced levels of several APOBEC3 proteins—including A3B, A3C, and A3G—in the presence of HHV-6B U28 (Fig. 4a, b), recapitulating the observations in HHV-6B-infected MT4 cells (Fig. 2c, d). Third, co-immunoprecipitation was performed in cells expressing StrepFlag–HHV-6B U28 and individual A3–AcGFPs in the presence of bafilomycin A1 (BafA1) to block lysosomal degradation. StrepFlag–HHV-6B U28 was pulled down using Strep-Tactin beads, and A3B– and A3C–AcGFP, but not A3H–AcGFP, were co-precipitated (Fig. 4c). These data indicate that HHV-6B U28 physically interacts with a subset of APOBEC3 proteins, promoting their cytoplasmic redistribution and/or degradation. For comparison, EBV BORF2 relocalizes A3A and A3B to the cytoplasm but does not alter their overall levels (Supplementary Fig. 4a–d), consistent with original reports[23,24]. Thus, HHV-6B U28 appears to counteract APOBEC3 proteins through a mechanism that is at least partly distinct from that employed by EBV BORF2.

Because HHV-6B U28 redistributed most APOBEC3 proteins except A3H (Fig. 3a), we next examined sequence determinants underlying this specificity. APOBEC3 proteins contain zinc-dependent catalytic domains (Z domains) categorized as Z1, Z2, or Z3 based on sequence features[30,31]. Human A3A and A3C each possess a single Z domain (Z1 or Z2, respectively), whereas A3B, A3D, A3F, and A3G harbor tandem Z domains—Z2-Z1 in A3B and A3G, and Z2-Z2 in A3D and A3F[30,31] (Fig. 3c). By contrast, A3H contains a single Z3 domain. Accordingly, we constructed A3C–A3H chimeras fused to AcGFP (Supplementary Fig. 5a) and co-expressed them with TagRFP–HHV-6B U28 in HEK293T cells. A chimera comprising the N-terminal half of A3C and the C-terminal half of A3H (A3CH-L4–AcGFP) colocalized with TagRFP–HHV-6B U28 in the cytoplasm, whereas the reciprocal chimera (A3HC-L4–AcGFP) did not (Supplementary Fig. 5b, c), indicating that the N-terminal half of A3C confers U28 recognition. We then generated A3H–AcGFP variants bearing A3C helix-1 plus loop-1[32,33] (A3CH-B1–AcGFP) or helix-1 alone (A3CH-L1–AcGFP). Strong cytoplasmic colocalization with TagRFP–HHV-6B U28 was observed for A3CH-B1–AcGFP, but was lost with A3CH-L1–AcGFP (Supplementary Fig. 5b, c). A series of A3A–A3H chimeras fused to AcGFP yielded similar outcomes (Supplementary Fig. 5d, e). Together, these data indicate that loop-1 residues of APOBEC3 proteins are key determinants for HHV-6B U28 recognition, whereas additional residues likely contribute to interactions that are required for degradation. In support of this interpretation, the loop-1 of A3H is structurally distinct from that of other APOBEC3 proteins (Supplementary Fig. 5f)[33].

Guided by the domain architecture of A3B (Z2 NTD and Z1 CTD), we next asked whether either domain alone is sufficient for HHV-6B U28–dependent relocalization. We constructed A3B NTD and CTD fusions with AcGFP (A3B-NTD–AcGFP and A3B-CTD–AcGFP, respectively) and co-expressed them with TagRFP–HHV-6B U28 in HEK293T cells. Co-expression with TagRFP–HHV-6B U28 elicited the characteristic cytoplasmic U28-positive domains, within which both A3B-NTD–AcGFP and A3B-CTD–AcGFP co-localized; however, the induction frequency and co-localization were lower for the CTD than for the NTD (Supplementary Fig. 6a, b). These results suggest that HHV-6B U28 recognizes both Z1 and Z2 domains but preferentially engages Z2 over Z1, in contrast to the EBV RNR (BORF2), which preferentially targets Z1 domains (e.g., in A3B and A3A)[23,25].

To further define the subcellular compartments in which HHV-6B U28 interacts with APOBEC3 proteins, we co-expressed A3C–AcGFP and Flag–HHV-6B U28 in HEK293T cells and performed

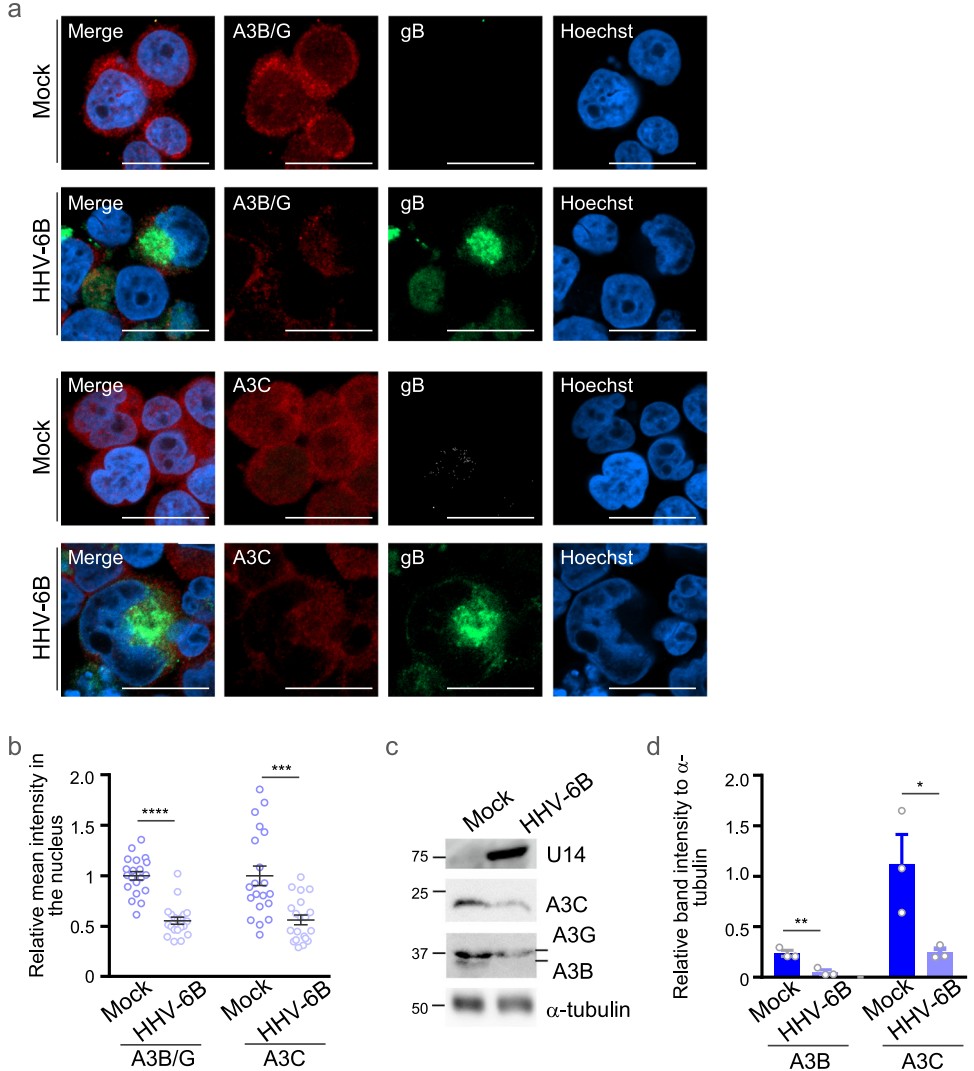

**Fig. 2 | Modulation of APOBEC3 protein localization and stability during HHV-6B infection. a** Immunofluorescence microscopy images of mock-infected or HHV-6B-infected MT4 cells for 72 h, stained with the indicated antibodies. Scale bars, 20 μm. Images are representative of three independent experiments. **b** Mean fluorescence intensity (MFI) of nuclear A3B/G or A3C. A3B/G denotes the signal detected with an antibody that recognizes both APOBEC3B and APOBEC3G and does not distinguish between them. Data points represent individual cells ($n = 20$ cells per condition) from one representative experiment and are shown as mean ± s.e.m. Statistical significance was assessed by unpaired two-tailed Student's t-test; ***$P < 0.001$; ****$P < 0.0001$. Similar results were obtained in three

independent experiments. The mean value in mock-infected cells was normalized to 100% relative MFI. **c** Immunoblots of lysates of MT4 cells infected with HHV-6B or mock-infected for 72 h. Immunoblot labels indicate molecular weight (kDa). **d** Intensities of A3B and A3C in each lane in (**c**), normalized to α-tubulin. In immunoblots, A3B and A3G were detected with an antibody that recognizes both proteins and were distinguished by their molecular weights. Data are shown as mean ± s.e.m. from three independent biological replicates ($n = 3$). Statistical significance was assessed by unpaired two-tailed Student's t-test; *$P < 0.05$; **$P < 0.01$. Exact $P$ values and detailed statistics for (**b**) and (**d**), and source data for (**b–d**) are provided in the Source Data.

immunofluorescence staining for organelle markers. The structures containing A3C–AcGFP and Flag–HHV-6B U28 showed partial colocalization with lysosomal (LAMP1) and autophagosomal (LC3B) markers (Fig. 5a, b). Although the endoplasmic reticulum marker calnexin was also diffusely distributed in the cytoplasm, its colocalization with Flag–HHV-6B U28 was weaker than that observed for LAMP1 and LC3B.

EBV BORF2 relocates A3B to cytoplasmic fiber-like structures. In parallel, the RNR large subunits of murine cytomegalovirus (MCMV; M45) and herpes simplex virus type 1 (HSV-1; ICP6) promote the aggregation of RIPK1, leading to its autophagic degradation as part of immune evasion strategies[34]. Similarly, HHV-6B U28 has been shown to aggregate NF-κB p65 without promoting its degradation[29]. To investigate the biophysical properties of A3C condensates induced by HHV-6B U28, we performed fluorescence recovery after photobleaching (FRAP). HEK293T cells were co-transfected with A3C–AcGFP and TagRFP–HHV-6B U28, and fluorescence recovery was monitored in live

cells after bleaching (Fig. 5c, d). While NF-κB p65–EGFP showed no recovery and NBR1–EGFP recovered within seconds, A3C–AcGFP fluorescence also recovered rapidly, suggesting that the U28-induced structures of A3C are liquid-like droplets rather than solid aggregates. Similarly, A3B–AcGFP domains induced by HHV-6B U28 were identified as liquid droplets, unlike the solid structures induced by EBV BORF2 (Fig. 5c, d). These results suggest that HHV-6B U28 sequesters APOBEC3 proteins in dynamic, non-aggregated compartments, representing a distinct evasion strategy from other herpesviruses.

## U28 protects the HHV-6B genome from APOBEC3 mutagenesis

Because HHV-6B resists the mutagenic activity of APOBEC3 proteins in MT4 cells (Fig. 1c), we investigated whether the HHV-6B U28 protein is required for maintaining viral genome integrity in these cells. MT4 cells were transduced with a lentiviral vector expressing *U28*-shRNA or

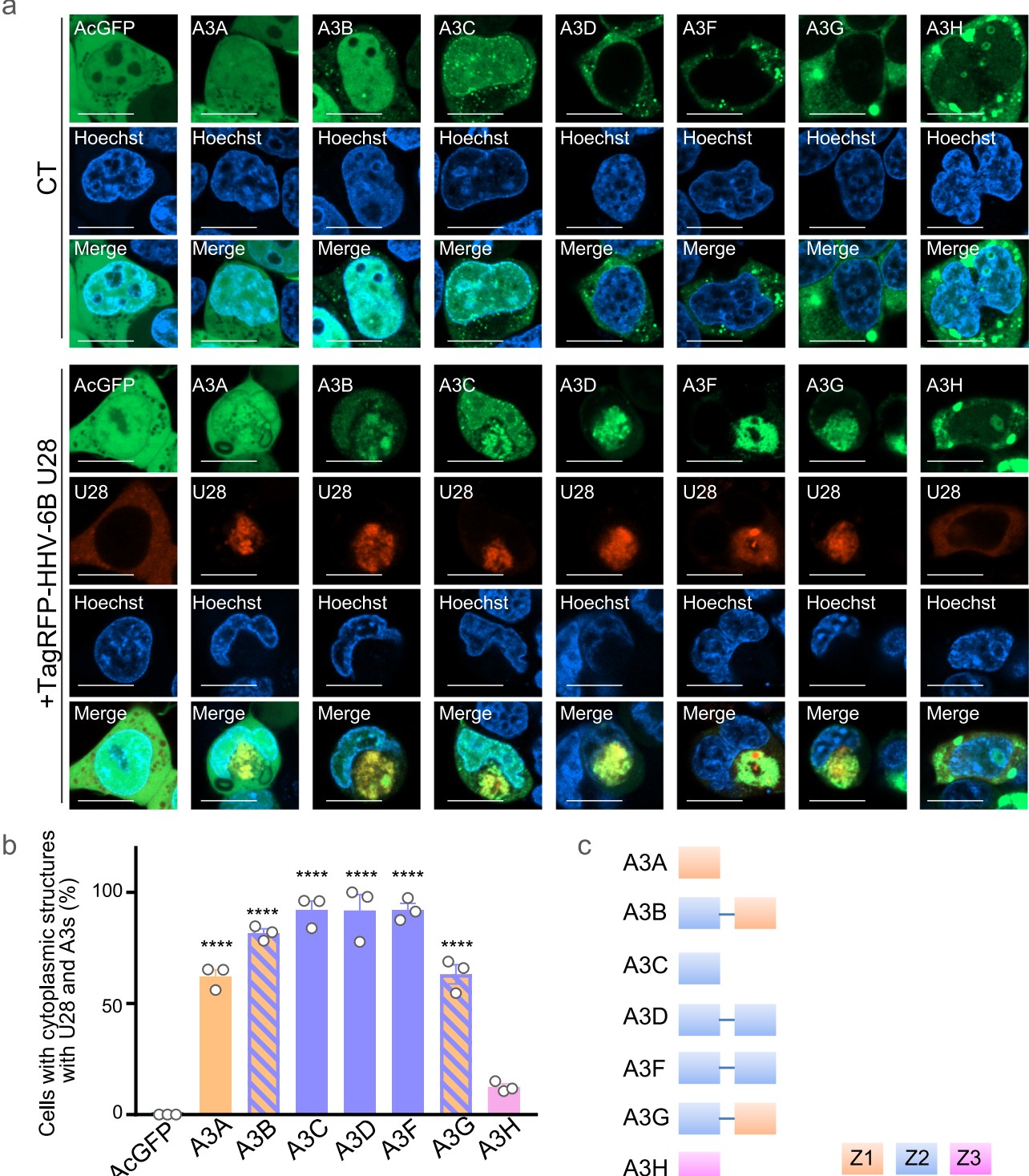

**Fig. 3 | Specificity of HHV-6B U28-induced relocalization among APOBEC3 proteins. a** Fluorescence microscopy images of HEK293T cells expressing individual APOBEC3 (A3)-AcGFP fusion proteins and TagRFP-tagged HHV-6B U28. CT, empty vector control. Scale bars, 10 μm. Images are representative of three independent experiments. **b** Percentage of cells with cytoplasmic A3-AcGFP and TagRFP-HHV-6B U28 in the experiments shown in (**a**). For each condition, 50–100 cells were analyzed per independent experiment. Data are shown as mean ± s.e.m. from three independent biological replicates ($n = 3$). Statistical significance was assessed by one-way ANOVA followed by Tukey's multiple-comparisons test; comparisons shown are relative to A3H; ****$P < 0.0001$. Exact $P$ values and detailed statistics for (**b**) are provided in the Source Data file. **c** Schematic diagram of seven human A3 proteins. A3 proteins possess a zinc-dependent catalytic domain (Z domain), designated Z1, Z2, or Z3 based on sequence features. Source data are provided in the Source Data file.

luciferase (Luc) controls (Supplementary Fig. 7a). Knockdown of *U28* restored A3B and A3C protein levels in HHV-6B-infected MT4 cells (Fig. 6a and b). Accordingly, *U28* knockdown also increased the frequency of C/G-to-T/A mutations without altering viral progeny copy numbers (Fig. 6c and Supplementary Fig. 7b). Consistent with APOBEC3 editing, motif analysis of these substitutions revealed enrichment in the TCW trinucleotide context with strand asymmetry and no enrichment at CpC motifs (Supplementary Fig. 7c), again arguing

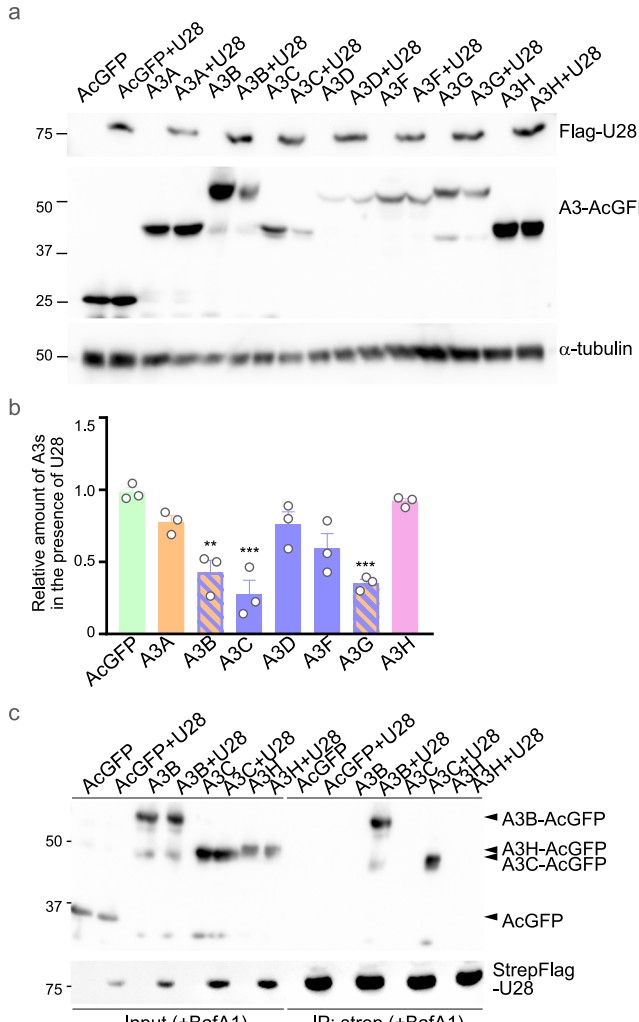

**Fig. 4 | Interaction of HHV-6B U28 with A3B and A3C and modulation of their protein levels. a** HEK293T cells were transfected with plasmids expressing individual APOBEC3 (A3)-AcGFP fusion proteins and Flag-tagged HHV-6B U28. At 48 h post-transfection, cells were analyzed by immunoblotting with the indicated antibodies. Immunoblot labels indicate molecular weight (kDa). Images are representative of three independent experiments. **b** Intensities of A3-AcGFP in the presence of HHV-6B U28 in (**a**), normalized to α-tubulin and divided by those without HHV-6B U28. Data are shown as mean ± s.e.m. from three independent biological replicates (*n* = 3). Statistical significance was assessed by one-way ANOVA followed by Tukey's multiple-comparisons test; comparisons shown are relative to A3H-AcGFP; **$P < 0.01$; ***$P < 0.001$. Exact *P* values and detailed statistics for (**b**) are provided in the Source Data file. **c** HEK293T cells were transfected with empty plasmid or plasmid expressing StrepFlag-tagged HHV-6B U28, together with A3B-, A3C-, or A3H-AcGFP plasmids, or with an AcGFP-expressing plasmid lacking an A3 fusion. At 46 h, cells were treated with 200 nM bafilomycin A1 (BafA1) for 2 h, then lysed in buffer. The lysates were precipitated with Strep-Tactin beads and analyzed by immunoblotting with the indicated antibodies. Immunoblot labels indicate molecular weight (kDa). Input whole-cell lysate, IP Strep-Tactin precipitate. Images are representative of three independent experiments. Source data are provided in the Source Data file.

against A3G as a significant contributor. These findings indicate that HHV-6B U28 promotes degradation of APOBEC3 proteins and thereby protects the viral genome from APOBEC3-mediated mutagenesis.

The open reading frames of *U28* from HHV-6A and HHV-6B share 99% sequence identity, and ectopic expression of Flag–HHV-6A U28 in HEK293T cells also relocalized and reduced the levels of several A3–AcGFP fusion proteins (Supplementary Fig. 4e–h). Nevertheless,

HHV-6A replication remains restricted in MT4 cells (Fig. 1e). We therefore examined *U28* expression and found that HHV-6B infection resulted in markedly higher *U28* mRNA levels than HHV-6A infection in MT4 cells (Fig. 6d). This difference in *U28* expression may underlie the divergent susceptibility of the two viruses to APOBEC3-mediated restriction. Supporting this idea, ectopic expression of either HHV-6A U28 or HHV-6B U28 significantly enhanced HHV-6A replication in MT4 cells (Fig. 6e and f). These results suggest that differential expression of *U28*, rather than differences in protein function, accounts for the distinct restriction phenotypes of HHV-6A and HHV-6B.

## HHV-6B but not HHV-6A genome integrity is maintained in vivo

To examine whether APOBEC3 proteins induce mutagenesis in HHV-6B genomes in vivo, we analyzed viral DNA from patients undergoing allo-HSCT, during which HHV-6B reactivation frequently occurs between 2 and 4 weeks post-transplant, with viral DNA detectable in plasma for extended periods (Supplementary Fig. 8a)[35]. The *U4* gene region analyzed is highly conserved among HHV-6B strains, with only one C/G-to-T/A variation identified across 37 publicly available genomes (Supplementary Table 1). Nevertheless, low-frequency C/G-to-T/A mutations were occasionally detected in the *U4* gene sequences derived from patient samples (Fig. 7a). A majority (60%) of the clones exhibited novel mutations relative to the HHV-6B HST reference genome, predominantly C/G-to-T/A substitutions (Supplementary Fig. 8b). Mutation rates varied between patients but correlated with the viral genome copy number in serum (Fig. 7b and Supplementary Fig. 8c). Motif analysis of patient-derived sequences showed no enrichment of TCW but a relative elevation at GpC (Supplementary Fig. 8d), indicating that canonical APOBEC3 footprints are not dominant at this locus in vivo. Given the high conservation of this locus among natural isolates, we consider these mutations likely to have arisen de novo during infection, rather than reflecting polymorphisms in circulating HHV-6B strains. Consistent with the lack of a TCW footprint, the integrity of the viral genome was largely preserved over periods exceeding 20 days. In agreement with this, such mutations were rarely detected in *U4* sequences derived from spleens of HHV-6B-infected humanized mice reconstituted with human immune cells[36] (Supplementary Fig. 8e, f). Together, these observations suggest that, in vivo, HHV-6B exhibits low-level substitutions that are not primarily driven by APOBEC3-type editing, and overall genome integrity is maintained.

To assess genome integrity during primary infection, we analyzed sera and cerebrospinal fluid (CSF) samples from three patients diagnosed with exanthema subitum-associated encephalopathy (Supplementary Fig. 9a). C/G-to-T/A mutations were also detected in the sera and CSF samples, although most clones were identical to the HHV-6B HST reference sequence (Fig. 7c and Supplementary Fig. 9b). Importantly, the mutations observed in HHV-6B genomes from both HSCT and encephalopathy patients were not found in any reported circulating HHV-6B strains, further supporting a de novo origin. Although some positions overlapped with variants observed in HHV-6B propagated in MT4-*U28* knockdown cells (Fig. 7d), the in vivo spectra showed no enrichment of TCW, suggesting shared sites of susceptibility that are not predominantly shaped by APOBEC3 activity. Collectively, these results indicate that HHV-6B resists APOBEC3-mediated mutagenesis in vivo, with overall genome integrity maintained across diverse clinical contexts.

In contrast, HHV-6A is generally asymptomatic, and clinical samples containing replicating HHV-6A are rarely available. Both HHV-6A and HHV-6B can integrate their genomes into host telomeres and be vertically transmitted as inherited chromosomally integrated HHV-6 (iciHHV-6)[6]. IciHHV-6 is found in approximately 1% of the human population, though its clinical significance remains unclear. However, reactivation of iciHHV-6A has been reported in a patient with X-linked severe combined immunodeficiency (X-SCID) who was hospitalized

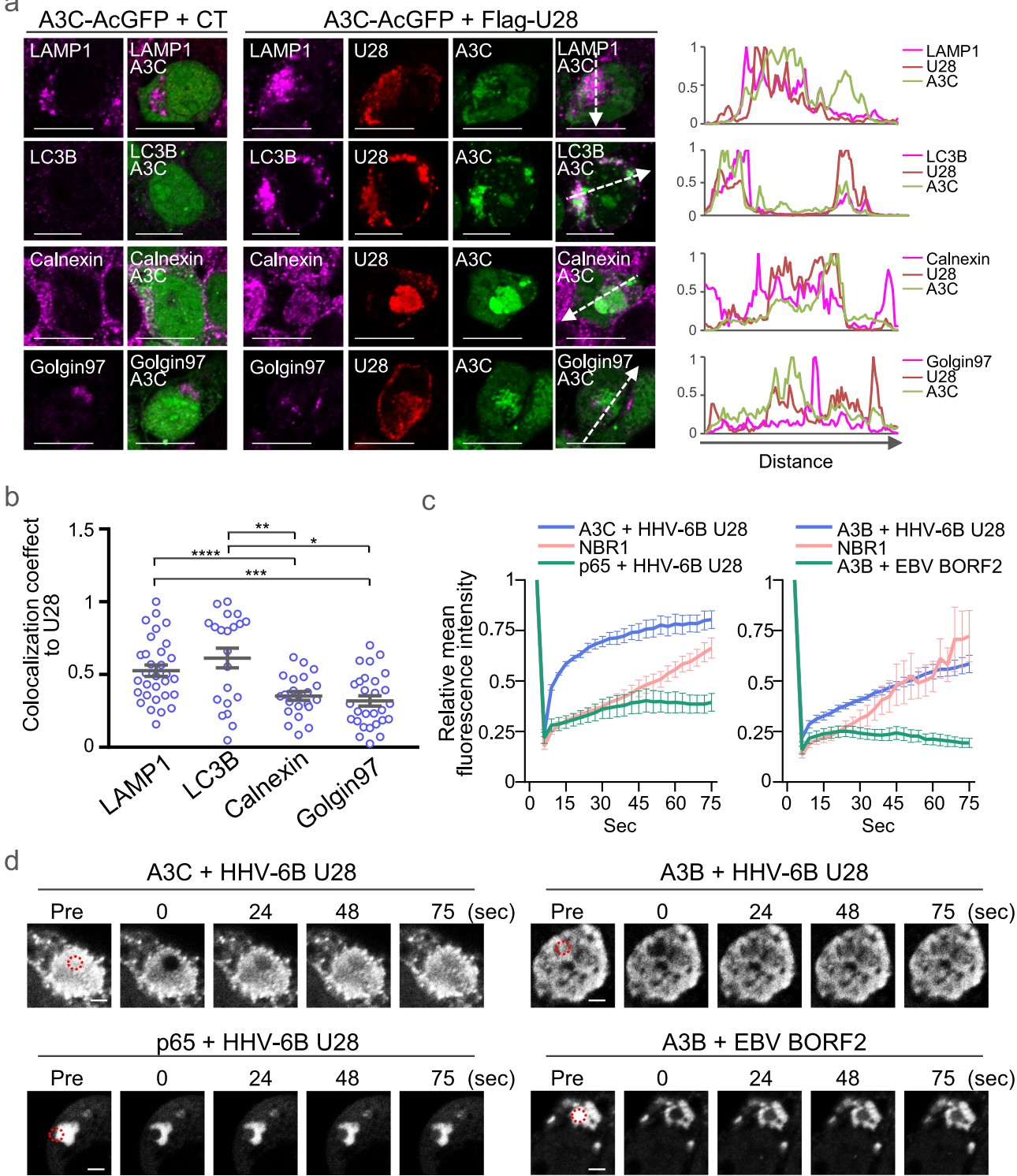

**Fig. 5 | Localization of APOBEC3 proteins to liquid-like cytoplasmic compartments by HHV-6B U28. a** Immunofluorescence microscopy images of HEK293T cells expressing A3C-AcGFP and control vector (CT) or Flag-HHV-6B U28, stained with the indicated antibodies. Scale bars, 10 μm. Images are representative of three independent experiments. Fluorescence line scans along the dotted lines in the fluorescence images are shown at the far right of each row. **b** Colocalization between Flag-HHV-6B U28 and LAMP1, LC3B, Calnexin, or Golgin97 was quantified using Manders' colocalization coefficient. Data points represent individual cells from one representative experiment (LAMP1, $n = 21$ cells; LC3B, $n = 31$ cells; Calnexin, $n = 28$ cells; Golgin97, $n = 22$ cells) and are shown as mean ± s.e.m. Statistical significance was determined by one-way ANOVA followed by Tukey's multiple-

comparisons test (*$P < 0.05$; **$P < 0.01$; ***$P < 0.001$; ****$P < 0.0001$). Similar results were obtained in three independent experiments. Exact $P$ values and detailed statistics for (**b**) are provided in the Source Data file. **c** HEK293T cells were transfected with the indicated plasmids for 48 h and analyzed by fluorescence recovery after photobleaching (FRAP). Mean fluorescence intensity of the bleached domains was quantified. Data points represent individual bleached domains from one representative experiment ($n = 10$ per condition) and are shown as mean ± s.e.m. Similar results were obtained in three independent experiments. EBV, Epstein–Barr virus. **d** Representative images from the experiment shown in (**c**). Red circles indicate the bleached area. Pre, prebleach. Scale bars, 2 μm. Source data for (**a**–**c**) are provided in the Source Data file.

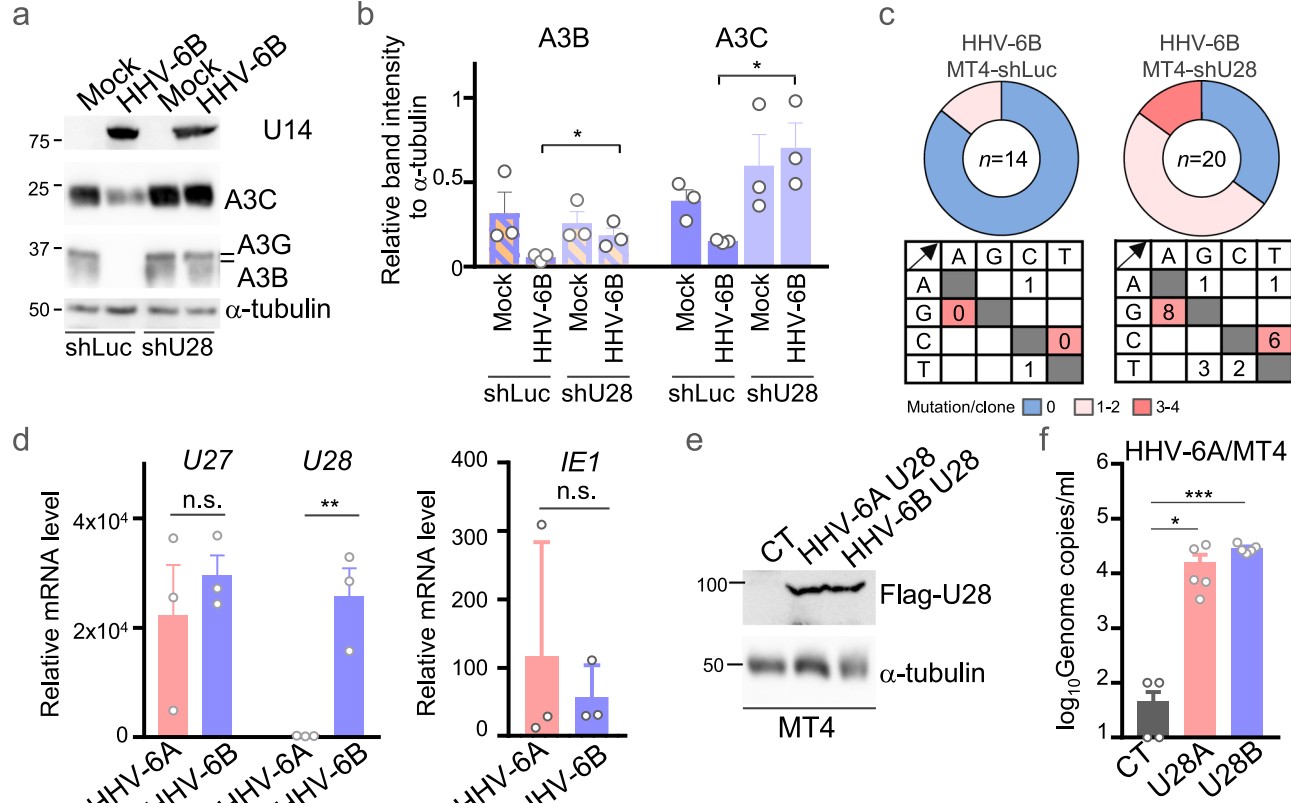

**Fig. 6 | Role of HHV-6 U28 in maintaining viral genome integrity. a** MT4 cells expressing shRNA to luciferase (shLuc) or *U28* (shU28) were infected with HHV-6B for 72 h and analyzed with the indicated antibodies. Immunoblot labels indicate molecular weight (kDa). Images are representative of three independent experiments. **b** Intensities of A3B or A3C in each lane in (**a**), normalized to α-tubulin. Data are shown as mean ± s.e.m. from three independent biological replicates (*n* = 3). Statistical significance was assessed by unpaired two-tailed Student's *t*-test; *P < 0.05. **c** Pie charts and mutation matrices showing mutational events in Sanger sequences of cloned *U4* gene segments from the supernatant of MT4 cells expressing the indicated shRNA and infected with HHV-6B for 72 h. Non-mutated sequences are depicted in blue; base substitutions with the number of mutations per sequence are depicted in red. **d** mRNA levels of the indicated viral genes by qPCR of MT4 cells infected with HHV-6A or HHV-6B for 48 h, normalized to β-actin.

Data are shown as mean ± s.e.m. from three independent biological replicates (*n* = 3). Statistical significance was assessed by unpaired two-tailed Student's *t*-test; n.s., not significant; **P < 0.01. **e** MT4-CT, MT4-HHV-6A U28 or MT4-HHV-6B U28 cells incubated with or without doxycycline (Dox) for 48 h were analyzed by immunoblotting. Immunoblot labels indicate molecular weight (kDa). Images are representative of three independent experiments. **f** Viral genome copy numbers in the supernatant of MT4 cells expressing HHV-6A U28 or HHV-6B U28 72 h after HHV-6A infection. Data are shown as mean ± s.e.m. from independent biological replicates (*n* = 5), where each *n* represents one independent infection experiment. Statistical significance was assessed by one-way ANOVA followed by Tukey's multiple-comparisons test; *P < 0.05; ***P < 0.001. Exact *P* values and detailed statistics for (**b**), (**d**), and (**f**) as well as source data for (**a**–**f**) are provided in the Source Data file.

for recurrent febrile episodes[37]. HHV-6A was isolated from the patient's peripheral blood mononuclear cells (PBMCs) during symptomatic periods, including 28 days after HSCT[37]. The isolated viruses were propagated in CBMCs, and their genomes were analyzed. No C/G-to-T/A mutations were detected in pre-HSCT viral genomes, whereas a high number of such mutations accumulated post-HSCT, with extensive sequence diversity (Fig. 8a, b, and Supplementary Fig. 9b). Importantly, these mutations were primarily present in viral DNA extracted directly from serum and plasma without in vitro culturing (Fig. 8a), and the overall diversity in HHV-6A was substantially higher than that observed in HHV-6B (Fig. 8b). Motif analysis showed a strand-dependent pattern: TpC was elevated on the plus strand, TCW enrichment was confined to the minus strand, and CpC was also elevated on the minus strand (Supplementary Fig. 9c). These features are compatible with APOBEC3-type editing but do not allow assignment to a single enzyme, suggesting mixed contributions and/or additional processes. As in HHV-6B, the same HHV-6A mutations were detected in both patient-derived and MT4-propagated viral genomes (Fig. 8c).

Consistent with these findings, *U28* mRNA expression in primary CBMCs infected in vitro with HHV-6B HST was significantly higher than in cells infected with HHV-6A GS, and C/G-to-T/A mutations

accumulated more prominently in the HHV-6A GS genome (Supplementary Fig. 10a, b). Furthermore, HHV-6B HST—but not HHV-6A GS—infection reduced APOBEC3 protein levels in these cells, and knockdown of *U28* increased HHV-6B genome mutation rates in primary CBMCs (Supplementary Fig. 10c, d). Collectively, these observations suggest that APOBEC3 proteins induce substantially more mutagenesis in HHV-6A than in HHV-6B, highlighting the role of differential *U28* expression in modulating viral genome integrity.

## Discussion

Herpesviruses are widely disseminated in nature, and nine species are known to primarily infect humans. Members of the family *Herpesviridae* share most essential genes and exhibit many conserved biological properties. However, the mechanisms underlying their divergent pathogenic profiles remain poorly understood. Our studies identify APOBEC3 proteins as factors that contribute to the differential susceptibility of HHV-6A and HHV-6B. We show that HHV-6B U28 redistributes APOBEC3 proteins to the cytoplasm and promotes their degradation, preserving viral genome integrity, whereas HHV-6A less efficiently counteracts APOBEC3 proteins (primarily A3B and A3C) and consequently accumulates higher mutation burdens.

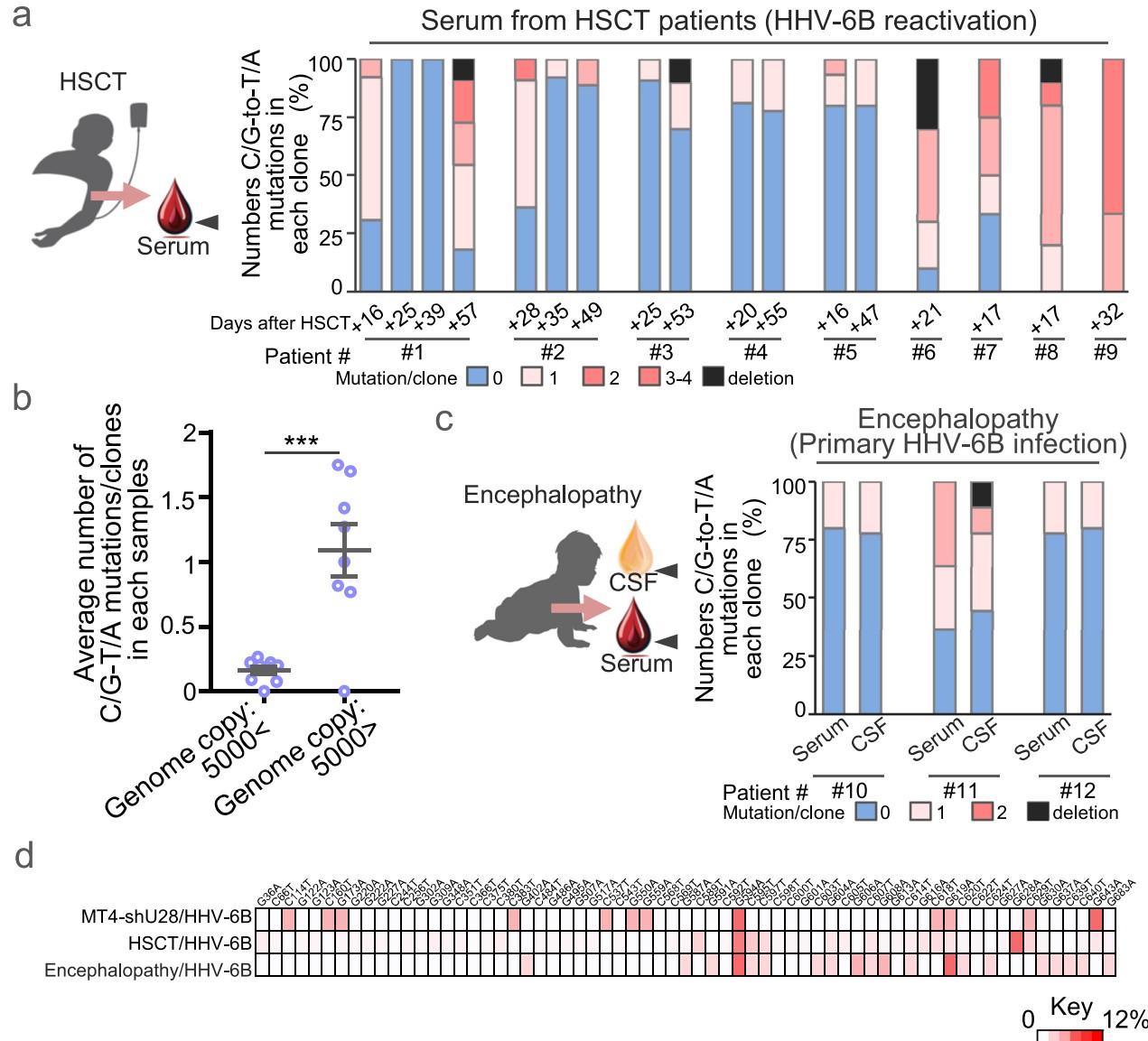

**Fig. 7 | In vivo analysis of APOBEC3-mediated mutations in HHV-6B genomes.** Stacked bar graphs showing the percentage of clones with the indicated number of C/G-to-T/A mutations in the *U4* gene segment, based on Sanger sequences from the serum of nine patients collected at the indicated days after hematopoietic stem cell transplantation (HSCT) (**a**), or from serum and cerebrospinal fluid (CSF) collected from three patients with exanthema subitum-associated encephalopathy (**c**). **b** Average number of C/G-to-T/A mutations per clone in each serum sample described in (**a**) was plotted separately according to genome copy number. Data points represent individual serum samples (low-copy group, *n* = 9 serum samples;

high-copy group, *n* = 8 serum samples) and are shown as mean ± s.e.m. Statistical significance was assessed by an unpaired two-tailed Student's *t*-test; ***P < 0.001. Exact P values and detailed statistics for (**b**) are provided in the Source Data file. **d** Heatmap representing the percentage of C/G-to-T/A mutations at the indicated positions in three sets of samples: HHV-6B genomes from the patients shown in panels (**a**) and (**c**), and HHV-6B HST genomes collected from MT4-shU28 cells (Fig. 6c). Distribution of mutation percentages is shown using the color scale below. Source data for **a**–**d** are provided in the Source Data file.

Among various APOBEC3 proteins, A3H was completely resistant to degradation by HHV-6B U28. Seven human *A3H* haplotypes have been described, but only three produce stable proteins[38,39]. Notably, stable *A3H* alleles are more frequent in individuals of African ancestry. As has been suggested for HIV-1[38,39], *A3H* genotype may also influence susceptibility to HHV-6B, as HHV-6B is not a predominant cause of roseola in Sub-Saharan African infants[29,30]. Further investigation is warranted to determine whether the prevalence of stable A3H haplotypes contributes to lower HHV-6B infection rates in these populations.

Although HHV-6B was largely protected from extensive mutagenesis in cell culture, we detected low-frequency C/G-to-T/A substitutions in clinical samples. Motif analysis of patient-derived sequences did not show enrichment of canonical APOBEC3 context (TCW) mutations and instead revealed a relative increase at GpC

motifs, indicating that a canonical TCW-type APOBEC3 footprint is not the dominant driver of in vivo variation at the *U4* locus. The high conservation of this region among circulating strains suggests that these substitutions arose de novo during infection rather than reflecting standing polymorphism; however, their spectra and sparsity are more consistent with non-APOBEC3 processes (for example, polymerase error, oxidative damage, repair, or recombination) together with selection and potential U28-mediated antagonism that prevents accumulation of heavily edited genomes. Mutation frequencies varied between patients and tracked with viral load, but our small cohort and heterogeneous clinical contexts preclude firm conclusions regarding determinants such as APOBEC3 expression or immune status. Overall, these findings support a model in which HHV-6B exhibits low-level, site-biased sequence change in vivo at the *U4* locus, with

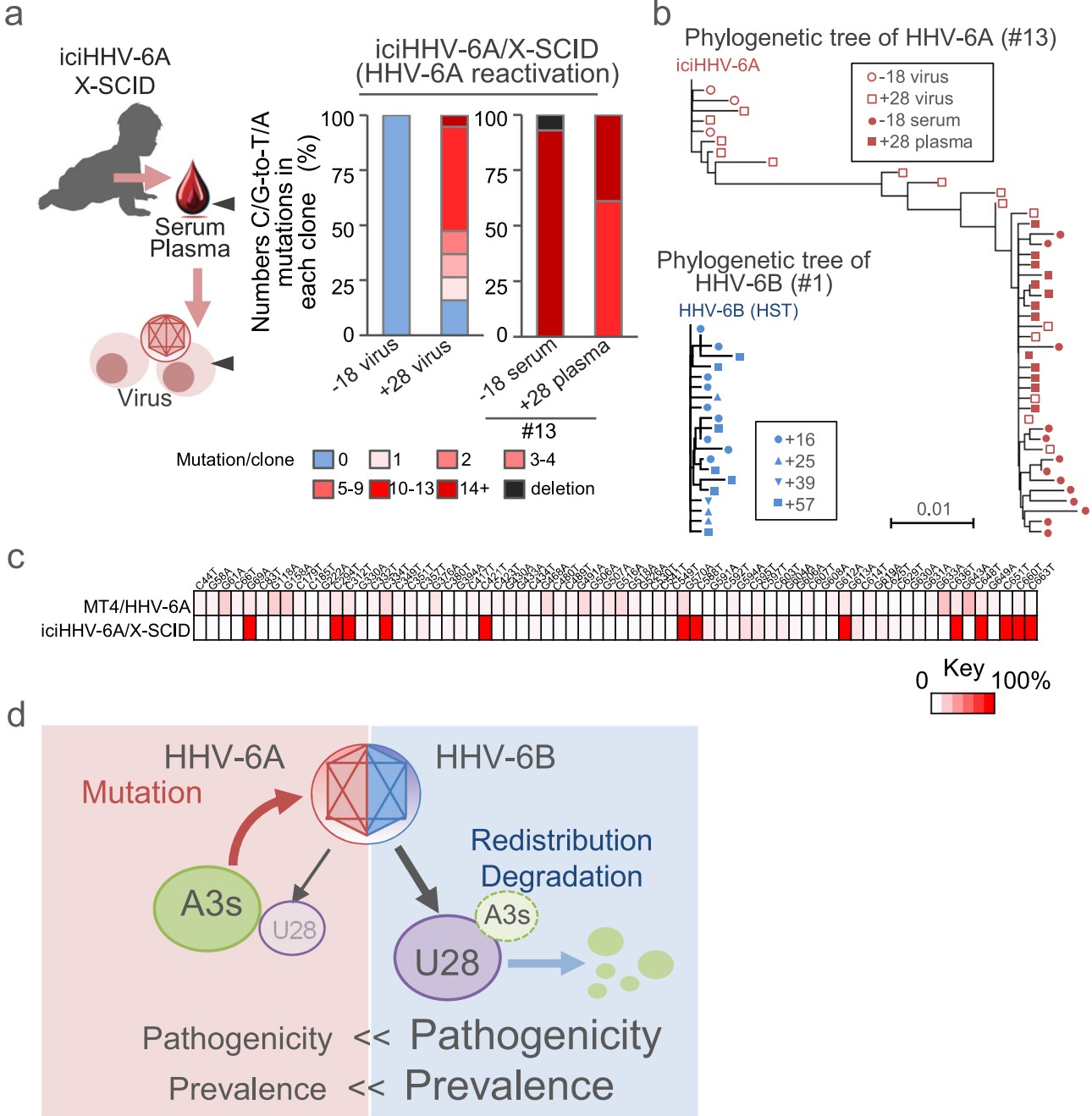

**Fig. 8 | In vivo analysis of APOBEC3-mediated mutation patterns in HHV-6A genomes. a** Stacked bar graphs showing the percentage of clones with the indicated number of C/G-to-T/A mutations in the *U4* gene segment, based on Sanger sequences of viral DNA collected either after in vitro propagation in cord blood mononuclear cells (CBMCs) infected with HHV-6A isolated from a patient with X-linked severe combined immunodeficiency (X-SCID) undergoing inherited chromosomally integrated HHV-6A (iciHHV-6A) reactivation (left), or directly from serum and plasma of the same patient without in vitro propagation (right). Serum and plasma were collected 18 days before and 28 days after transplant, respectively; -18 virus and +28 virus indicate HHV-6A isolates propagated in CBMCs from samples collected at these time points, whereas -18 serum and +28 plasma indicate viral DNA analyzed directly from the corresponding clinical specimens.
**b** Dendrograms showing alignment of mutated *U4* segments from patients 1 and 13, as described in Supplementary Figs. 8a and 9a. **c** Heatmap representing the percentage of C/G-to-T/A mutations at the indicated positions in two sets of samples: HHV-6A genomes from the patient shown in (**a**), and HHV-6A GS genomes collected from MT4 cells (Fig. 1c). Distribution of mutation percentages is shown using the color scale below. **d** Overview of the effect of APOBEC3 (A3) proteins on the two closely related viruses, HHV-6A and HHV-6B. Source data for (**a**–**c**) are provided in the Source Data file.

limited evidence of canonical TCW-type APOBEC3 editing, while multiple mechanisms beyond APOBEC3 likely contribute to the observed variation, as described for HSV-1 and HSV-2[40,41].

HHV-6A clinical isolates are now exceedingly rare. Our analysis of HHV-6A from a single patient revealed a higher mutational burden than in any HHV-6B sample, consistent with failure to evade APOBEC3-mediated restriction. This may contribute to the difficulty in isolating replication-competent HHV-6A today. Given that APOBEC3-induced mutations can impair genome integrity and viral fitness, inability to neutralize these enzymes likely limits HHV-6A replication, persistence, and transmission.

In this study, we highlight A3C because, despite its relatively low catalytic activity[42,43], knockout experiments demonstrated that it restricts HHV-6A replication and promotes mutagenesis. We

prioritized A3C as an experimentally tractable model—its smaller size and single Z2 domain enabled clean domain swaps and chimeras—through which we mapped U28 recognition to the N-terminal loop-1 (Supplementary Fig. 5), accounting for A3H (Z3) resistance and the susceptibility of Z1/Z2 APOBEC3 proteins, including A3B. Importantly, U28 antagonizes both A3C and A3B, and given A3B's stronger catalytic activity on nuclear DNA, A3B may represent an even more significant restriction factor for HHV-6 in vivo. Our findings are in line with prior studies reporting mutagenic roles of A3C in tumors[44,45] and in herpesvirus genomes[46], supporting the biological relevance of A3C-mediated editing. Thus, our choice to pursue APOBEC3 proteins was guided by mechanistic rationale and subsequent functional validation, rather than DEG p-values per se.

Across the Herpesviridae, several RNR homologs—especially in α- and γ-herpesviruses—can antagonize APOBEC3 proteins, with virus- or subfamily-specific target preferences (Supplementary Fig. 4 and refs. 23,47). However, this is not universal: β-herpesviruses such as HCMV reportedly do not employ the RNR to neutralize APOBEC3 enzymes[28]. HHV-6B U28 is notable for counteracting a broad range of APOBEC3 proteins, which may be critical given the high APOBEC3 expression in hematopoietic cells[32,33]. In contrast, HHV-6A expresses lower levels of U28. Although their *U28* coding and upstream regions are largely conserved, scattered substitutions and genomic overlap with *U29* may influence *U28* transcription, complicating functional dissection. Notably, *U28* expression differences are not due to general defects in early or immediate early gene expression, as *U27* and *IE1* levels are comparable between HHV-6A and HHV-6B. It is possible that HHV-6A has evolved toward a more latent or less cytopathic lifecycle, in which strong U28 expression is not selected. These differences in U28 may help explain the greater pathogenic potential and prevalence of HHV-6B relative to HHV-6A. In addition, iciHHV-6B, but not iciHHV-6A, has been linked to increased risk of autoimmune diseases such as systemic lupus erythematosus (SLE)[48]. One possible explanation is that HHV-6B more efficiently evades APOBEC3 restriction, enabling stable replication and persistence of its genome. However, whether disease phenotypes associated with iciHHV-6B require active viral replication remains unclear and will require further investigation.

It is possible that HHV-6A replication is restricted to host cells with inherently low APOBEC3 expression. This raises the possibility that APOBEC3 genes have acted as selective pressures driving divergence between HHV-6A and HHV-6B following the split of the human and chimpanzee lineages[5,49]. In chimpanzees, A3C forms dimers and shows higher activity than in humans, whereas most humans carry a low-activity variant and only a minority harbor the S188I polymorphism (~10% of people of African descent), which confers higher activity through a distinct dimer interface[50]. Such differences may in part have influenced the divergence of HHV-6A, which is less resistant to A3C, and HHV-6B, which is more resistant, in humans. More broadly, the ongoing evolutionary arms race between viruses and intrinsic antiviral defenses may have contributed to the emergence of the nine human herpesviruses.

Our study focused on the *U4* region, selected for its high mutation susceptibility identified in pilot 3D-PCR screens. While this provided valuable initial insights, broader patterns of mutagenesis may exist elsewhere in the viral genome. Future work incorporating high-throughput sequencing will be essential to characterize genome-wide mutational contexts and assess selective pressures on non-coding regions, as demonstrated in recent studies of DNA viruses like mpox[51]. Together, these findings provide a framework for understanding how APOBEC3 enzymes contribute to herpesvirus genome evolution and pathogenesis.

## Methods

### Cells
MT4, HSB2, and JJhan cells maintained in the laboratory of Yasuko Mori were cultured in RPMI 1640 medium containing 8% fetal bovine serum (FBS)[11,52]. Umbilical cord blood mononuclear cells (CBMCs) were cultured in RPMI 1640 medium containing 8% FBS[52,53]. CBMCs were purchased from the Cell Bank of the RIKEN BioResource Center, Tsukuba, Japan. The use of CBMCs in this study was approved by the Ethics Committee of Kobe University Graduate School of Medicine (approval number: No. 1209). HEK293T cells maintained in the laboratory of Yasuko Mori were cultured in DMEM containing 8% FBS. The human cell lines used in this study were checked against the current ICLAC Register of Misidentified Cell Lines, and none were listed. The authors did not further authenticate the cell lines after receipt. Transfection experiments were performed using Lipofectamine 3000 (Thermo Fisher Scientific, Waltham, MA)[54–56].

### Viruses
HHV-6A strain GS and HHV-6B strain HST were propagated in activated CBMCs until cytopathic effects were maximal. Cells were lysed by a single freeze-thaw cycle at −80 °C. Cell debris was removed by centrifugation at 1500×g for 5 min. Supernatants were used for virus infections[11,52,53]. HHV-6A GS and HHV-6B HST were used to infect HSB2 or MT4 cells[11,52,53]. Briefly, $1 \times 10^6$ cells were collected by centrifugation, resuspended in medium containing $1 \times 10^3$ (HHV-6A) or $2 \times 10^6$ (HHV-6B) genome copies of the virus, or medium alone for mock infection. For CBMC experiments, $1 \times 10^5$ genome copies of HHV-6A or HHV-6B were used. Cells were centrifuged at 35 °C for 30 min at 1700×g, then cultured in RPMI with 2% FBS. Infection efficiency was ~40–50% at 48 h, determined by immunofluorescence staining with antibodies against gQ1 or gB. At 72 h post-infection, supernatants were harvested, and viral DNA was extracted using DNeasy Blood & Tissue Kits (Qiagen). Viral genome copies per mL were quantified by qPCR using SYBR Select Master Mix.

### Plasmids and oligonucleotides
lentiCRISPR-A3B and lentiCRISPR-A3C were constructed by inserting annealed oligonucleotides (Supplementary Table 3) into the Esp3I site of lentiCRISPR v2 (Addgene #52961)[57]. pInd10-Luc and pInd10-U28 encode shRNAs targeting luciferase and *U28*, respectively, under a Dox-inducible promoter. pcDNA3.1-Flag-HHV-6A U28 and pcDNA3.1-Flag-HHV-6B U28 encode Flag-tagged HHV-6A U28 and HHV-6B U28, respectively[29]. pcDNA3.1-BORF2-HA was a gift from T. Murata[58]. TagRFP was amplified from pTagRFP-C (Evrogen, FP141) and cloned into the U28 or BORF2 plasmids to create pTagRFP-HHV-6A U28 and pTagRFP-EBV BORF2. StrepFlag-HHV-6B U28 and pTagRFP-HHV-6B U28 have been reported[29]. For pInd20-HHV-6A U28 and pInd20-HHV-6B U28, the Flag-HHV-6A U28 and Flag-HHV-6B U28 ORFs were subcloned into pInducer20 (Addgene #44012)[59]. Plasmids encoding AcGFP-A3A to -A3H were constructed by inserting CBMC cDNA or pCMV-hA3B-BE3 (Addgene plasmid #113411)[60] into pAcGFP-N1.

Plasmids pAcGFP-A3CH-L4, A3HC-L4, A3CH-L1, and A3CH-B1, carrying the fusion proteins of the A3C-A3H chimeras with AcGFP, were constructed by amplifying from pAcGFP-A3C or pAcGFP-A3H and cloning into pAcGFP-N1 (Clontech, 632469). A3CH-L4, A3HC-L4, A3CH-L1 or A3CH-B1 are chimeras of A3C (codons 1–79) and A3H (codons 68–200), A3H (codons 1–67) and A3C (codons 80–190), A3C (codons 1–22) and A3H (codons 15–200), A3C (codons 1–31) and A3H (codons 29–200), respectively (Supplementary Fig. 5a). Plasmids pAcGFP-A3AH-B1 and A3AH-L1, carrying the fusion proteins of the A3A-A3H chimeras with AcGFP, were constructed by amplifying from pAcGFP-A3A or pAcGFP-A3H and cloning into pAcGFP-N1. A3AH-B1 or A3AH-L1 are chimeras of A3A (codons 1–29) and A3H (codons 29–200) or A3A (codons 1–23) and A3H (codons 15–200), respectively (Supplementary Fig. 5a). Plasmids pAcGFP-A3B-NTD and pAcGFP-A3B-CTD, encoding AcGFP fusions with A3B (codons 1–193) and A3B (codons 187–382), respectively, were generated by PCR amplification from pAcGFP-A3B and subcloning into pAcGFP-N1.

All oligonucleotide sequences used in this study, including primers, shRNA sequences, and CRISPR guide sequences, are listed in Supplementary Table 3. Newly generated plasmids and other unique reagents are available from the corresponding author upon reasonable request.

## Construction of cell lines

HEK293T cells were transfected with lentiCRISPR constructs plus packaging plasmids pCAG-HIV-gp and pCMV-VSV-G-RSV-Rev[53,61,62]. At 48 h, supernatants were used to transduce MT4 cells, selected with 1 μg/mL puromycin. MT4-derived lines included MT4-casCT, -casA3B, -casA3C, -casA3C, -HHV-6A U28, -HHV-6B U28, -shLuc, and -shU28. For induction, cells were treated with 1 μg/mL Dox for 24 h. CBMCs were transduced with pInd10 constructs and cultured with puromycin, 2 ng/mL of IL-2 and 5 ng/mL of phytohemagglutinin for 7 days.

## Antibodies and reagents

For immunoblotting and immunofluorescence analysis, we used mouse monoclonal antibodies (mAb) against Flag (Sigma-Aldrich, cat. no. F3165, clone M2, 1:1000), HA (Sigma-Aldrich, cat. no. H9658, clone HA-7, 1:1000), α-tubulin (Sigma-Aldrich, cat. no. CP06, clone DM1A, 1:1000), LAMP1 (Santa Cruz Biotechnology, cat. no. sc-20011, clone H4A3, 1:200) and Golgin97 (Thermo Fisher Scientific, cat. no. 14-9767-82, clone CDF4, 1:200), and rabbit mAb against APOBEC3B (generated by the R.S.H. laboratory, 5210-87-13[63], 1:500), and rabbit polyclonal antibodies against GFP (Abcam, cat. no. ab290, 1:1000), Flag (MBL, cat. no. PM020, 1:200), APOBEC3C (GeneTex, cat. no. GTX102164, 1:500), Calnexin (Abcam, cat. no. ab22595, 1:500) and LC3B (Abcam, cat. no. ab51520, 1:500). The mAbs to HHV-6A IE2 (AIE2-1), U14 (BU14), gB (OHV-1) and gQ1 (AgQ1-119) were produced from hybridomas and their supernatants were used directly as described[64–67]. Antibodies to U14 and gB were used for infection marker in immunoblotting and immunofluorescence, respectively. The lysosomal inhibitor bafilomycin A1 (BafA1) was purchased from Selleck Chemicals.

## Affinity precipitation

HEK293T cells ($9 \times 10^5$) were transfected with plasmids expressing A3-AcGFP and StrepFlag-HHV-6B U28. At 46 h, cells were treated with 200 nM BafA1 for 2 h, then lysed in buffer (0.5% NP-40, 150 mM NaCl, 50 mM Tris−HCl, pH 8.0). Lysates were clarified by centrifugation and incubated with Strep-Tactin beads (IBA) for 2 h at 4 °C. Beads were washed and analyzed by immunoblotting.

## Immunoblotting

Cells were lysed with SDS sample buffer (62.5 mM Tris−HCl [pH 6.8], 20% glycerol, 2% sodium dodecyl sulfate [SDS], 50 mM dithiothreitol) and electrophoresed in denaturing gels, and then transferred to nitrocellulose or polyvinylidene difluoride membranes. The membranes were blocked with 5% skimmed milk in PBS-T (PBS containing 0.05% Tween 20) for 30 min and reacted with the indicated antibodies for at least 1 h at room temperature or 4 °C. The membranes were then reacted with secondary antibodies conjugated with peroxidase (Cytiva) and visualized using ECL with Fusion FX6.EDGE (Vilber Bio Imaging). The intensities of the indicated bands were evaluated using ImageJ software and normalized to those of α-tubulin.

To analyze the reduction of APOBEC3 proteins in transfected cells, $9 \times 10^5$ HEK293T cells were co-transfected with the plasmid expressing individual A3-AcGFP together with Flag-HHV-6A U28, Flag-HHV-6B U28, or EBV BORF2-HA plasmids. At 48 h post-transfection, the cells were analyzed by immunoblotting. To analyze the reduction of the quantity of APOBEC3 proteins in infected cells, $1 \times 10^6$ MT4 cells were infected with HHV-6B HST. At 72 h post-infection, the infected cells were analyzed by immunoblotting. Infection was confirmed using mAb against HHV-6B U14 protein[53]. Uncropped and unprocessed scans are provided in the Source Data file.

## Immunofluorescent microscopy

HEK293T cells were transfected with the indicated plasmids using Lipofectamine 3000 for 48 h. The cells were fixed with 4% paraformaldehyde for 10 min, permeabilized with 0.1% Triton-X100 for 10 min, and blocked with PBS containing 0.4% bovine serum albumin (BSA) for 30 min. These cells were stained with the indicated antibodies for 1 h at room temperature, followed by secondary antibodies conjugated to Alexa Fluor (Thermo Fisher Scientific) for 1 h at room temperature. Finally, the cells were inspected using an LSM800 microscope (Zeiss)[54,55,68,69]. Images acquired by the LSM800 microscope were analyzed with the colocalization function in ZEN3.1 software (Zeiss). Regions of the image that did not contain a visual signal were selected and used to threshold the image. Manders' colocalization coefficient ($M$) was calculated using the following equation, as described previously[70,71]. $M = \Sigma_i X_{i.\text{colocalized}} / \Sigma_i X_i$, where $X_i$ is equal to the intensity of marker $X$ at a pixel and $X_{i.\text{colocalized}}$ is the intensity of the pixels where the intensity of the other marker is greater than the threshold value. An $M$ value of 1.0 indicates 100% colocalization, and 0 indicates 0% colocalization.

To determine viral infectivity, the infected cells were fixed with methanol/acetone and stained with mAb against IE2, U14, and/or gQ1. Nuclear DNA was stained with Hoechst 33342 and specific signals were detected using a fluorescence microscope (BZ-X810, Keyence)[53].

## FRAP assay

HEK293T cells transfected with A3-AcGFP, EGFP-p65, or EGFP-NBR1 plus TagRFP-HHV-6B U28 or TagRFP-EBV BORF2 were used. Domains were bleached with 75% laser intensity at 488 nm for 0.5 s, and recovery was monitored at 3 s intervals.

## Reverse transcription quantitative PCR (RT-qPCR)

One million cells were infected with either HHV-6A GS or HHV-6B HST or treated with medium. At 48 h post-infection, total RNA was isolated from the cells using NucleoSpin RNA kits (Macherey-Nagel) according to the manufacturer's instructions, and cDNA was synthesized from the isolated RNA with SuperScript III (ThermoFisher Scientific). The amount of cDNA of specific genes (*IE1*, *U27*, *U28*, and β-actin) was quantified using the SYBR Select master mix (Thermo Fisher Scientific) according to the manufacturer's instructions. The amount of these mRNAs was normalized to the amount of β-actin mRNA. The relative abundance of each gene product was calculated using the comparative cycle threshold ($2^{-\Delta\Delta CT}$) method[52,53].

## Mutational analysis

Viral DNA was extracted from the supernatant of the infected cells using the DNeasy blood and tissue kit (QIAGEN). HHV-6 *U4* genes were amplified with primers U4 OUT-F and U4 OUT-R, followed by secondary PCR with primers U4 IN-F and U4 IN-R (Supplementary Table 3). The second PCR fragments (684 bp) were then cloned into pCR4-TOPO vectors (Thermo Fisher Scientific) or pBlueScript KS+ (Stratagene), and the indicated number of successful recombinant clones was selected randomly and sequenced[72]. The *U4* gene of HHV-6A GS (GenBank accession number: KC465951.1) or HHV-6B HST (GenBank accession number: AB021506.1) was used as a reference sequence to determine the presence of mutations.

To evaluate APOBEC3-associated mutation signatures, trinucleotide contexts centered on each mutated site were extracted from the reference sequences. For each clone, the mutated nucleotide and its immediate upstream base were classified into NpC categories (TpC, CpC, GpC, or ApC). The observed frequencies of these contexts were compared with their expected frequencies based on the availability of each NpC in the corresponding reference sequence. In addition, mutations occurring in TC dinucleotides were further divided into TCW (W = A or T) and TC−nonW categories. For this analysis, the

expected proportion of TCW was calculated relative to all NpC motifs in the reference sequence.

For the mutational analysis in mice, viral DNA was collected from the spleens of HHV-6B HST-infected humanized mice engrafted with human immune cells, as described in Wang et al. [36]. The samples used in this study were remaining materials from the previously published work, and no new animal experiments were performed. Samples were from 6 mice, designated 51, 53, 55, b72, b81, and r28, sacrificed 6–9 days after infection. All animal experiments in the original study were approved by the Institutional Animal Care and Use Committee and were handled in accordance with Kobe University animal experimentation regulations (permit numbers 1209 and 160162).

### Mutational analysis of clinical samples

For the mutational analysis in patients 1–9 described in Supplementary Fig. 8a, viral DNA was collected from the serum of patients who underwent allo-HSCT at Kobe University Hospital between August 2016 and May 2017, as described in Nagamata et al.[73]. All patients gave informed consent before the collection of samples, and this study was approved by the Ethics Committee of Kobe University Graduate School of Medicine (approval number: No. 1567). Participants received no compensation. The *U4* gene of HHV-6B HST was used as a reference sequence to determine the presence of mutations.

For the mutational analysis in patients 10–12 described in Supplementary Fig. 9a, viral DNA was collected from serum and cerebrospinal fluid (CSF) of patients with exanthema subitum-associated acute encephalopathy at the time of admission to Fujita Health University Hospital between March 2010 and September 2011. For the mutational analysis in patient 13 described in Supplementary Fig. 9a and reported by Endo et al. [37], viral DNA was collected from serum or plasma at the indicated time. Isolated viruses from the patient's peripheral blood mononuclear cells at the same time[37] were propagated in CBMCs and collected as described above. The *IE1* gene sequence was analyzed with primers PO3IE1A and PO3IE1C[74] (Supplementary Table 3) to confirm the HHV-6A genome but not the HHV-6B genome. Informed consent was obtained from the patient's guardians. This study was approved by the Ethical Review Board of Human Studies at Fujita Health University (approval number: EN23-118) and the Ethics Committee of Kobe University Graduate School of Medicine (approval number: B230111). In the case of patient 13, the *U4* gene of iciHHV-6A (Supplementary Table 2) was used as a reference sequence to determine the presence of mutations.

### Sequence alignments

Sequence alignments were analyzed with MEGA 11 software to construct phylogenetic trees by the Neighbor-joining method. To analyze conservation of the HHV-6A and HHV-6B *U4* gene in strains, the PCR-amplified region of the *U4* gene of HHV-6A GS or HHV-6B HST was compared with the corresponding region of 21 HHV-6A or 36 HHV-6B strains obtained from NCBI, coupled with the patient 13 iciHHV-6A (Supplementary Tables 1 and 2).

### RNA-seq library preparation and sequencing

Total RNAs of CBMCs, MT4, HSB2, or JJhan cells were isolated from the cells using NucleoSpin RNA kits (Macherey-Nagel) according to the manufacturer's instructions. Sequencing libraries were generated using NEBNext Poly(A) mRNA Magnetic Isolation Module and NEBNext UltraTMII Directional RNA Library Prep Kits (NEB) following the manufacturer's recommendations. After cluster generation, the library preparations were sequenced on a NovaSeq 6000[53]. Differentially expressed genes (DEGs) were identified using DESeq2 with the Wald test by comparing gene expression levels between the combined group of JJhan and HSB2 cells and the combined group of MT4 cells and CBMCs. Since this analysis was conducted for exploratory purposes, we applied a threshold of |log$_2$ fold change| >1 and an unadjusted *p*-value < 0.05. In Fig. 1b, only genes associated with GO:0045087 (innate immune response) were selected and displayed. DEGs were used for hypothesis generation and did not prespecify candidates for functional follow-up.

### Statistical analysis

For the comparison of two groups, statistical analysis was performed using the unpaired Student's *t*-test. Tukey's test was used for multiple comparisons. A *P* value > 0.05 was considered not significant (n.s.). All statistical analysis was performed using GraphPad Prism 7 (GraphPad Software). The data were assumed to meet the assumptions of parametric tests based on experimental design and distribution characteristics. No formal tests for normality or equal variance were performed. For context enrichment analysis of mutations, binomial tests were performed to compare observed versus expected frequencies of NpC categories or TCW motifs, where expected values were derived from their availability in the reference sequence. Multiple testing correction was applied using the Benjamini–Hochberg method, with NpC category tests evaluated at an FDR-adjusted $q < 0.05$ ($q \geq 0.05$ = n.s.) and the single-motif TCW test evaluated at $p < 0.05$ ($p \geq 0.05$ = n.s.).

### Sex and gender reporting

Sex was considered for the clinical materials used in this study and is reported in Supplementary Figs. 8a and 9a. Gender information was not available to the authors. No sex- or gender-based analysis was performed because the study was not designed or powered for such comparisons. For immortalized cell lines (HEK293T, MT4, HSB2, and JJhan), sex was not considered as an experimental variable and was not relevant to the study aims. For humanized mice, sex was not used for randomization or stratification. These points are detailed in the Nature Portfolio Reporting Summary.

### Reporting summary

Further information on research design is available in the Nature Portfolio Reporting Summary linked to this article.

## Data availability

The RNA-seq data generated in this study have been deposited in the DNA Data Bank of Japan under accession code DRA015097. Source Data are provided with this paper. All other data supporting the findings of this study are available within the article and its Supplementary Information. Source data are provided with this paper.

## Code availability

No custom code was used in this study.

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

## Acknowledgements

We thank Y. Ueda for the excellent technical assistance, F. Zhang, S. Elledge, J. Chen and T. Murata for kindly providing reagents, and the lab members for helpful comments. This study was supported by Grants for Scientific Research and Grant-in-Aid for Scientific Research from the Japan Society for the Promotion of Science (JSPS), 20H03496 (J.A.) and 23K18589 (J.A.); MEXT Leading Initiative for Excellent Young Researchers Grant (J.A.); SPRING, Japan Science and Technology Agency (JST), JPMJSP2148 (M.H.); contract research funds from the Japan Program for Infectious Diseases Research and Infrastructure, Japan Agency for Medical Research and Development (AMED), 20wm0325005h (J.A.); Precursory Research for Innovative Medical Care (PRIME), AMED, 22gm6410022h (J.A.); and grants from the Takeda Science Foundation (J.A.), MSD Life Science Foundation (J.A.), Shionogi Infectious Disease Research Promotion Foundation (J.A.), The Waksman Foundation of Japan (J.A.), and The Chemo-Sero-Therapeutic Research Institute (J.A.). S.A. was supported by a research fellowship from MEXT. Virus restriction studies in the Harris lab are supported in part by NIAID AI064046 (R.S.H.). R.S.H. is an Investigator of the Howard Hughes Medical Institute, a CPRIT Established Investigator (RR220053), and the Ewing Halsell President's Council Distinguished Chair at the University of Texas San Antonio.

## Author contributions

Conceptualization: J.A. and S.A.; Methodology: Y.K., H.M., B.W., S.N., and M.N.; Investigation: J.A., S.A., J.R.H., and M.H.; Supervision: J.A., T.Y., R.S.H., and Y.M.; Writing—original draft: J.A., S.A.; Writing—review and editing: J.A., R.S.H.

## Competing interests

The authors declare no competing interests.
