## [Peer Review file · Nature Communications]

A viral APOBEC3 antagonist distinguishes HHV-6A from HHV-6B

Corresponding Author: Dr Jun Arii

Version 0:

Reviewer comments:

Reviewer #1

(Remarks to the Author)

This is a very interesting study and the authors have addressed the concerns raised in the previous round of review.

Minor comments

1. Line 305 and elsewhere. The authors' focus on A3C seems curious. Unless, I missed something, the evidence for A3B as the driver of the mutagenesis seems just as good, and given its stronger activity on DNA in the nucleus, probably better. Do the mutational signatures help identify which A3 protein is the driver?
2. The authors might mention, that like A3H, there are polymorphisms in A3C in other species and in humans with increased activity due to increased dimer formation, e.g. DOI: 10.1093/nar/gkx066 . I presume that the less active form is present in the cell lines and models used here? In any case, this might be important for their discussion in line 330 on the divergence of HHV6 from the common ancestor of humans and chimps, and well as influence HHV6 in different populations.
3. Figure 1D is quite ugly. As the signal looks strong, the authors should be able to produce a better blot.

Reviewer #2

(Remarks to the Author)

In this revised manuscript, Arii et al show that HHV6A and 6B are mutated at the U4 locus. They attribute these mutations to APOBEC enzymes, which are differentially expressed to some extent in permissive v. non-permissive cells. The manuscript reports that the HHV6 U28 protein both degrades APOBEC enzymes and re-localizes them to the cytosol to mitigate interaction with the virus. Correlative analysis of samples taken from infected patients indicate C>T mutations within HHV6 genomes which the authors interpret as being caused by APOBEC activity on viral genomes.

My overall assessment is that this is an interesting start, but much of the data presented is fairly minimal in comparison to the interpretations. Each figure shows just 1-2 experiments (and several analyses of that experiment) that forms the basis for a substantial biological claim. Further rigor of experiments, controls, and additional concordant data are required to fully support the interpretations stated by the authors.

Major points for consideration:

1. Overall, the data presented are not of high quality. Western blots do not have even loading controls, IF images show oddly shaped cells and inconsistent nuclear staining. Quantifications of these raw data are fine and statistics are appreciated, but the blots and images are not strong enough to independently make the points regarding mislocalization, colocalization, liquid-like cystolic structures, and interacting proteins. Some specific examples are:
 - a. Fig 2A – even uninfected cells have very odd shape/nuclear staining. Some images appear to show multiple cells with contradictory results (HHV 6B infected cells show both cytosolic and nuclear A3G/B). Are there better representative images? At the very least, more cells with consistent findings should be shown.

- b. Nearly all loading controls on western blots are uneven or oversaturated, which degrades confidence in the data
- c. Why are input and IP on different gels in Fig 4C?
- 2. Fig 3A – are the authors concluding that every single A3 enzyme re-localized by U28? This would be an unusual finding given the specificity of A3 enzymes for their viral targets. In other words, it is unprecedented in the literature to find that every single A3 targets the same virus or viral protein. Thus, this point deserves further explanation/investigation.
- 3. The authors claim that A3 is degraded by U28, which is based on several western blots in Fig 4. The input blot in 4C shows that A3 levels are consistent across U28-transfected lysates and A3-only controls. This is inconsistent with the decreased A3 levels found in U28-transfected lysates in panel A.
- 4. Sequencing interpretations in Figs 1, 7, and 8 are correlative. The claim that C>T mutations are caused by APOBEC would be strengthened by showing that these C>T mutations occur in a TC context, or in clusters, or at putative hairpins, all of which would be consistent with what has previously been demonstrated as APOBEC activity.

Minor suggestions:

- 1. The rationale for further investigation of A3 proteins remains weak – A3B is not statistically significant in the DEG analysis (Fig 1B) though it is pursued as an HHV6 restriction factor in subsequent figures. Are other genes in the DEG analysis known to be HHV6 restriction factors that can be pointed to as controls for the experiment?
- 2. Fig 1F would benefit from a casCT control analysis.

Version 1:

Reviewer comments:

Reviewer #1

(Remarks to the Author)

The authors have addressed all of my comments from the previous round of review.

Reviewer #2

(Remarks to the Author)

The authors have been very responsive to reviewers' comments and their efforts are commendable. Some of the data still fall short of interpretations, but in this revised version data presented are of higher quality and more robust. It remains unclear whether the mutations detected in HHV6 genomes in experimental systems and in patients are caused by APOBEC. Some of the new included analyses shed light on this remaining question, for example the authors detect TCW context mutations in experimental systems but more variable mutational patterns in vivo and in patients.

Version 2:

Reviewer comments:

Reviewer #3

(Remarks to the Author)

As requested, my review focuses on whether the study's claims are supported by the data and whether the experimental approaches are in line with current standards in the APOBEC-virus field.

Overall speaking, the manuscript presents compelling biochemical, cellular, and genetic evidence that APOBEC3 proteins restrict HHV6A replication and that HHV6B U28 counteracts this restriction. The genetic data (A3B/A3C knockout) and mutation signature analyses strongly support APOBEC3 mediated restriction of HHV6A. I think the conclusion could be even more solid if the U28 deletion HHV6B virus (similar to Vif deletion in HIV study for A3s) can be produced to further validate U28 function. But given the siRNA result, this may not be easily done, and is not an essential piece of evidence for the claims the authors have derived.

Point-by point Response to the Reviewers

Reviewer #1:

General comments: This is a very interesting study and the authors have addressed the concerns raised in the previous round of review.

Response: We sincerely thank the reviewer for the encouraging overall assessment of our work. We greatly appreciate the recognition that this is an interesting study and that the revisions have addressed the concerns from the previous review round. This feedback is highly motivating and has helped us to further strengthen the manuscript.

Comment 1: (i) Line 305 and elsewhere. The authors' focus on A3C seems curious. Unless, I missed something, the evidence for A3B as the driver of the mutagenesis seems just as good, and given its stronger activity on DNA in the nucleus, probably better.

(ii) Do the mutational signatures help identify which A3 protein is the driver?

Response: (i) We focused on A3C not to downplay A3B, but because A3C provided a tractable system to map viral antagonism at residue-level resolution. As newly shown in Extended Data Fig. 5, A3C/A3H/A3A chimeras delimit loop-1 in the N-terminal domain as the determinant for U28 recognition, and swapping this element is sufficient to switch sensitivity to U28-mediated relocalization without reducing protein accumulation. This mechanistic handle would have been far more difficult with A3B's two-domain (Z2–Z1) architecture. Importantly, the loop-1 rule rationalizes why A3H (Z3) is refractory, while Z1/Z2 A3s (including A3B) are susceptible—thus insights gained with A3C generalize to the A3B clade. Consistent with this, U28 relocalizes and reduces both A3C and A3B (Fig. 2–4), and A3C or A3B knockout increases HHV-6A replication with a corresponding drop in C/G-to-T/A mutagenesis (Fig. 1e, f). Furthermore, guided by this model, we performed domain-sufficiency assays for A3B: AcGFP fusions of the Z2 NTD and the Z1 CTD were each recruited to U28-positive cytoplasmic compartments upon co-expression with TagRFP–HHV-6B U28, with the NTD showing higher recruitment efficiency (Extended Data Fig. 6a, b). We therefore now state explicitly that A3B likely exerts an even stronger *in vivo* effect, while A3C serves as the mechanistic proxy that allowed us to identify the transferable recognition element (loop-1) (Page 21. Lines 356–368).

(ii) Thank you for the question. As suggested, we performed additional mutational signature analyses on the sequencing datasets. In multiple datasets, we now observe a clear enrichment of TpC and TCW contexts accompanied by strand asymmetry, which is characteristic of APOBEC3-associated signatures and allows us to rule out A3G (no

enrichment at CpC motifs) (Extended Data Fig. 2b, 7c, 9c). However, these context/strand features are shared to varying degrees by A3A, A3B, and A3C, and thus do not permit unambiguous discrimination among A3A/B/C at present; we therefore refrain from assigning a single driver and explicitly note this as a limitation and a priority for future work (e.g., targeted perturbation or catalytic-dead rescue under the sequencing conditions).

By contrast, in patient-derived HHV-6B sequences, although C/G-to-T/A substitutions were relatively frequent, neither TpC nor TCW enrichment was detected, indicating that canonical APOBEC3 footprints are not dominant in vivo at this locus and suggesting that HHV-6B may evade A3-mediated editing in patients (Extended Data Fig. 8d). We have incorporated these findings into the Results (Pages 6-7, lines 113-117; page 14, lines 233-238; pages 15-16, lines 264-271; page 18, lines 302-306), figure legends, and Discussion (Page 20, lines 336-350) to clarify where APOBEC3-like signatures are evident and where they are not.

Comment 2: The authors might mention, that like A3H, there are polymorphisms in A3C in other species and in humans with increased activity due to increased dimer formation, e.g. DOI: 10.1093/nar/gkx066 . I presume that the less active form is present in the cell lines and models used here? In any case, this might be important for their discussion in line 330 on the divergence of HHV6 from the common ancestor of humans and chimps, and well as influence HHV6 in different populations.

Response: We thank the reviewer for this insightful comment. As suggested, we have revised the Discussion (Page 23, lines 391-396) to acknowledge A3C polymorphisms and their potential impact on HHV-6 divergence. Specifically, we now note that in chimpanzees A3C forms dimers and shows higher activity than in humans, whereas most humans carry the low-activity form and only a minority harbor the high-activity S188I polymorphism (~10% of people of African descent), which confers increased activity through a distinct dimer interface (DOI: 10.1093/nar/gkx066). We further state that such differences in A3C activity may in part have influenced the divergence of HHV-6A and HHV-6B in humans.

Comment 3: Figure 1D is quite ugly. As the signal looks strong, the authors should be able to produce a better blot.

Response: We thank the reviewer for this valuable suggestion. In response, we repeated the experiment and present an improved blot in the revised Fig. 1d, where the strong signal is now more clearly and cleanly visualized. We believe this presentation better

supports the conclusion.

Reviewer #2:**General points**

In this revised manuscript, Arii et al show that HHV6A and 6B are mutated at the U4 locus. They attribute these mutations to APOBEC enzymes, which are differentially expressed to some extent in permissive v. non-permissive cells. The manuscript reports that the HHV6 U28 protein both degrades APOBEC enzymes and re-localizes them to the cytosol to mitigate interaction with the virus. Correlative analysis of samples taken from infected patients indicate C>T mutations within HHV6 genomes which the authors interpret as being caused by APOBEC activity on viral genomes.

My overall assessment is that this is an interesting start, but much of the data presented is fairly minimal in comparison to the interpretations. Each figure shows just 1-2 experiments (and several analyses of that experiment) that forms the basis for a substantial biological claim. Further rigor of experiments, controls, and additional concordant data are required to fully support the interpretations stated by the authors.

Response: We thank the reviewer for the careful reading of our revised manuscript and for summarizing the scope of our study. We appreciate the constructive feedback regarding the depth of evidence relative to the interpretations. While we are encouraged that the reviewer finds the study interesting, we agree that additional rigor and data will further strengthen our conclusions. Accordingly, we revised the manuscript to (i) improve data presentation (updated representative IF images and evenly loaded blots; input and IP shown on the same gel with +BafA1 indicated), (ii) clarify mechanism and scope (U28 acts broadly on Z1/Z2 but not Z3; new A3B domain-sufficiency data showing stronger recruitment of the Z2 NTD; Extended Data Fig. 6a, b), (iii) add trinucleotide-context and strand-asymmetry analyses where appropriate and temper claims where APOBEC signatures are not predominant, and (iv) include additional controls and textual clarifications (e.g., casCT in Fig. 1f; exploratory nature of the RNA-seq screen). We also edited the text to state limitations explicitly and to indicate where future work is needed.

Major points for consideration:

Comment 1: Overall, the data presented are not of high quality. Western blots do not have even loading controls, IF images show oddly shaped cells and inconsistent nuclear staining. Quantifications of these raw data are fine and statistics are appreciated, but the blots and images are not strong enough to independently make the points regarding mislocalization, colocalization, liquid-like cystolic structures, and interacting proteins. Some specific examples are:

a. Fig 2A – even uninfected cells have very odd shape/nuclear staining. Some images

appear to show multiple cells with contradictory results (HHV 6B infected cells show both cytosolic and nuclear A3G/B). Are there better representative images? At the very least, more cells with consistent findings should be shown.

b. Nearly all loading controls on western blots are uneven or oversaturated, which degrades confidence in the data

c. Why are input and IP on different gels in Fig 4C?

Response: We thank the reviewer for these constructive comments. In response, we have carefully re-examined our raw data and replaced the relevant figures with improved versions. Specifically:

a. Figure 2a has been replaced with representative images from additional experiments, showing consistent nuclear morphology and clearer patterns of A3 localization.

b. Western blots in Fig. 1d, 2c, 4a, 4c, 6a and 6e, and Extended Data Fig. 4c and 4g have been replaced with higher-quality blots using evenly loaded controls, avoiding oversaturation and ensuring more reliable presentation.

c. Figure 4c has been revised so that the input and IP are shown on the same gel, improving clarity and comparability.

We believe that these revisions strengthen the presentation of our data and more clearly support the conclusions.

Comment 2: Fig 3A – are the authors concluding that every single A3 enzyme re-localized by U28? This would be an unusual finding given the specificity of A3 enzymes for their viral targets. In other words, it is unprecedented in the literature to find that every single A3 targets the same virus or viral protein. Thus, this point deserves further explanation/investigation.

Response: Thank you for this insightful comment. We are not concluding that U28 re-localizes every APOBEC3 enzyme. Our data indicate that U28 broadly redirects members of the Z1 and Z2 clades (for example, A3A, A3B, A3C, A3F, A3G) to the cytoplasmic domain, whereas the Z3 enzyme A3H is an exception and is not re-localized by U28. To clarify scope and mechanism, we generated chimeras between U28-responsive A3s and A3H and found that introducing a short N-terminal domain segment comprising helix 1 plus loop 1 from A3C into A3H conferred U28-dependent recruitment to the cytoplasmic domain; an A3A (Z1)–A3H chimera engineered with the same N-terminal segment produced an equivalent gain of function. These results suggest that U28 recognizes features near loop 1 within the N-terminal domain that are shared among Z1 and Z2, providing a parsimonious explanation for broad yet clade-selective relocalization. In

agreement with this notion, loop-1 of A3H is structurally distinct from the other A3s (Bohn et al. Nat Commun 2017, PMID: 29044109). Consistent with Z2-biased recognition, domain-sufficiency assays using A3B showed that both the Z2 NTD and the Z1 CTD fusions were recruited by U28, with the NTD exhibiting stronger co-localization than the CTD (Extended Data Fig. 6a, b). We have revised the Results to state explicitly that U28 acts broadly on Z1/Z2 but not on Z3 (Pages 10-12, lines 174-203), and we have expanded the Discussion to propose this helix-1–loop-1–centered recognition model (Pages 21-22; lines 357-372).

Comment 3: The authors claim that A3 is degraded by U28, which is based on several western blots in Fig 4. The input blot in 4C shows that A3 levels are consistent across U28-transfected lysates and A3-only controls. This is inconsistent with the decreased A3 levels found in U28-transfected lysates in panel A.

Response: Thank you very much for this important and thoughtful comment. We agree that, as presented, panels Fig. 4a and 4c could appear inconsistent. The apparent discrepancy reflects a deliberate difference in experimental conditions. In Fig. 4a we examined steady-state A3 levels without inhibitors, where U28 expression was associated with a decrease in A3 signal. By contrast, Fig. 4c was performed in the presence of bafilomycin A1 (BafA1) to block lysosomal degradation, ensuring that sufficient A3 remained for the co-IP/interaction assay. Under this stabilized condition, A3 levels appear comparable between U28-transfected and control lysates. To prevent confusion, we have revised the Results text to state explicitly that BafA1 was added in Fig. 4c to inhibit degradation (Page 10, lines 165-166), updated the Fig. 4c legend and labels to indicate “+BafA1,” and added a cross-reference to the Materials and Methods, where the BafA1 treatment was already described. These clarifications align the interpretation of Fig. 4a, where degradation is observed at steady state, and Fig. 4c, where degradation is suppressed to permit assessment of protein complexes.

Comment 4: Sequencing interpretations in Figs 1, 7, and 8 are correlative. The claim that C>T mutations are caused by APOBEC would be strengthened by showing that these C>T mutations occur in a TC context, or in clusters, or at putative hairpins, all of which would be consistent with what has previously been demonstrated as APOBEC activity.

Response: Thank you for this constructive suggestion. In response, we re-analyzed the sequencing datasets underlying Figs. 1, 6, 7, and 8 to evaluate trinucleotide context and strand asymmetry—hallmarks of APOBEC3 mutagenesis. In HHV-6A–infected MT4 cells (Fig. 1c), substitutions were enriched in the TCW context and displayed strand

asymmetry (Extended Data Fig. 2b), consistent with APOBEC-associated signatures (SBS2); because context alone cannot distinguish A3A from A3B or A3C, we avoid assigning a single enzyme, although the paucity of 5'-CC motifs argues against A3G (Pages 6-7, lines 112-117).

In the HHV-6B shU28 condition (Fig. 6b), motif analysis likewise revealed TCW enrichment with strand asymmetry and no enrichment at CpC (Extended Data Fig. 7c), again supporting APOBEC3-type editing and arguing against A3G as a major contributor (Page 14, lines 233-238).

By contrast, in patient-derived HHV-6B sequences (Fig. 7a), we did not observe enrichment of TCW/TpC contexts; rather, substitutions were relatively elevated at GpC (Extended Data Fig. 8d), indicating that canonical APOBEC3 footprints are not dominant at this locus in vivo (Pages 15-16, lines 264-267; pages 16-17, lines 282-287).

In patient-derived HHV-6A sequences (Fig. 8a), we detected a strand-dependent pattern—TpC increased on the plus strand, TCW enrichment confined to the minus strand, and CpC also elevated on the minus strand (Extended Data Fig. 9c)—which is compatible with APOBEC-type editing yet does not permit attribution to a single enzyme, suggesting mixed contributions and/or additional processes. (Pages 18, lines 302-306) We have revised the Results and figure legends accordingly and adjusted the Discussion to indicate where the data match SBS2-like features and where APOBEC3 signatures are not predominant (Page 20, lines 336-350).

Minor suggestions:

Minor Comment 1: The rationale for further investigation of A3 proteins remains weak – A3B is not statistically significant in the DEG analysis (Fig 1B) though it is pursued as an HHV6 restriction factor in subsequent figures. Are other genes in the DEG analysis known to be HHV6 restriction factors that can be pointed to as controls for the experiment?

Response: We appreciate the reviewer's comment. As suggested, we clarified in the Results that other differentially expressed genes with potential antiviral functions were also identified. Specifically, we added: "The analysis identified multiple differentially expressed genes with potential roles in herpesvirus replication or restriction, such as RELB, REL, and AIM2, which warrant future investigation." These examples illustrate that genes with reported or plausible roles in herpesvirus restriction were present among the DEGs.

Importantly, our decision to pursue APOBEC3 genes was not based on single-gene DEG significance. The RNA-seq screen was exploratory (Methods), and we prioritized the

APOBEC3 family based on family-level patterns (an overall rightward shift of several members in Fig. 1b) rather than single-gene p-values (Page 5, lines 83-91). This prioritization was then tested functionally and independently of expression statistics: (i) A3B/A3C knockout increased HHV-6A replication and reduced C/G-to-T/A mutagenesis (Fig. 1e, f); (ii) HHV-6B U28 relocalized and decreased A3 proteins (Figs. 2, 4); and (iii) mutation contexts were APOBEC-consistent in HHV-6A but not in HHV-6B in vivo (Figs. 1c, 6, 7). Thus, APOBEC3 genes were advanced based on mechanistic validation, whereas other DEGs (e.g., RELB, REL, AIM2) are noted as hypothesis-generating and will be pursued in future work.

To prevent over-interpretation of the DEG screen, we also updated the Methods to state that the analysis was exploratory and that “DEGs were used for hypothesis generation and did not prespecify candidates for functional follow-up.” (Page 36, lines 597-598). In the Discussion, we further note that our choice to pursue APOBEC3 genes was guided by mechanistic rationale and subsequent functional validation, not by DEG p-values per se (Page 21, lines 365-368).

Minor Comment 2: Fig 1F would benefit from a casCT control analysis.

Response: We thank the reviewer for this helpful suggestion. In response, we have added the casCT control analysis to Fig. 1f. The revised figure now includes this control, which confirms that the observed effects are specific and not due to Cas9 expression itself.

We thank the reviewers for their thoughtful evaluation and constructive comments. We have resubmitted a revised manuscript with tracked changes, together with this point-by-point response. In this appeal submission, our revisions are limited to a targeted clarification in the Discussion to constrain the scope of conclusions drawn from in vivo and patient-derived sequence observations (no new data).

Reviewer #1

Comment: The authors have addressed all of my comments from the previous round of review.

Response: We thank Reviewer #1 for their positive assessment and for confirming that all prior concerns were addressed. No additional changes were requested.

Reviewer #2

Comment: The authors have been very responsive to reviewers' comments and their efforts are commendable. Some of the data still fall short of interpretations, but in this revised version data presented are of higher quality and more robust. It remains unclear whether the mutations detected in HHV6 genomes in experimental systems and in patients are caused by APOBEC. Some of the new included analyses shed light on this remaining question, for example the authors detect TCW context mutations in experimental systems but more variable mutational patterns in vivo and in patients.

Response: We thank Reviewer #2 for recognizing the improved robustness of the revised manuscript. We respectfully submit that APOBEC3 causality is established in the controlled infection setting by APOBEC3 genetics: mutation accumulation during HHV-6A infection is strongly reduced or abrogated in APOBEC3B- and APOBEC3C-deficient engineered cells (Fig. 1f), and depletion of APOBEC3B or APOBEC3C enhances HHV-6A replication (Fig. 1e), demonstrating a direct fitness impact.

Regarding heterogeneous patterns in vivo and in patient-derived sequences, we agree that such patterns should be interpreted cautiously and need not be uniform even if APOBEC3

contributes. Importantly, the manuscript provides a mechanistic basis for non-uniform clinical patterns: HHV-6B encodes U28-mediated APOBEC3 evasion, and substitutions increase when this antagonism is impaired (Fig. 6c; Extended Data Fig. 10d), consistent with APOBEC3 pressure being counteracted to different degrees between HHV-6A and HHV-6B in clinical contexts.

To prevent any overinterpretation from the patient-derived sequence observations, we have revised the Discussion to explicitly constrain our *in vivo* interpretation to the sequenced U4 locus and to canonical TCW-type footprint inference, without changing any underlying data or adding new experiments.

Change made in manuscript: Discussion section, paragraph on HHV-6B clinical samples—concluding interpretation and scope revised to (i) explicitly state “at the U4 locus” and (ii) specify “limited evidence of canonical TCW-type APOBEC3 editing,” thereby avoiding genome-wide inference from locus-level observations. (Pages 20-21, lines 340-356)